# Ribosome collisions trigger cis-acting feedback inhibition of translation initiation

Szymon Juszkiewicz, Greg Slodkowicz[†], Zhewang Lin[†], Paula Freire-Pritchett, Sew-Yeu Peak-Chew, Ramanujan S Hegde*

MRC Laboratory of Molecular Biology, Francis Crick Avenue, Cambridge, United Kingdom

**Abstract** Translation of aberrant mRNAs can cause ribosomes to stall, leading to collisions with trailing ribosomes. Collided ribosomes are specifically recognised by ZNF598 to initiate protein and mRNA quality control pathways. Here we found using quantitative proteomics of collided ribosomes that EDF1 is a ZNF598-independent sensor of ribosome collisions. EDF1 stabilises GIGYF2 at collisions to inhibit translation initiation in cis via 4EHP. The GIGYF2 axis acts independently of the ZNF598 axis, but each pathway's output is more pronounced without the other. We propose that the widely conserved and highly abundant EDF1 monitors the transcriptome for excessive ribosome density, then triggers a GIGYF2-mediated response to locally and temporarily reduce ribosome loading. Only when collisions persist is translation abandoned to initiate ZNF598-dependent quality control. This tiered response to ribosome collisions would allow cells to dynamically tune translation rates while ensuring fidelity of the resulting protein products.

*For correspondence:
rhegde@mrc-lmb.cam.ac.uk

[†]These authors contributed equally to this work

## Introduction

Rapidly growing cells typically produce an average of one to two proteins per ribosome per minute (*Princiotta et al., 2003*). Both the amount and quality of proteins produced from a cell's transcriptome are crucial for cellular homeostasis (*Labbadia and Morimoto, 2015*). Thus, cells have evolved numerous mechanisms for regulating translation output (*Roux and Topisirovic, 2018*) and monitoring protein quality during and after translation (*Wolff et al., 2014*). One of the earliest monitors of protein quality detects ribosomes that stall during translation (*Brandman and Hegde, 2016*; *Joazeiro, 2019*). A ribosome that slows excessively can be an indicator of several potential problems including a damaged or incorrectly processed mRNA (*Shoemaker and Green, 2012*), amino acid or energy insufficiency (*Harding et al., 2019*; *Inglis et al., 2019*), or certain types of cellular stress (*Liu et al., 2013*; *Shalgi et al., 2013*). Inability to respond appropriately to these problems leads to disrupted protein homeostasis and disease (*Kapur and Ackerman, 2018*).

Because a common cause of ribosome stalling is a defective mRNA, stalled ribosome detection is linked to mRNA decay (*Shoemaker and Green, 2012*) and degradation of the partially synthesised protein (*Bengtson and Joazeiro, 2010*; *Joazeiro, 2019*). There are at least two mechanisms that cells use to detect a stalled ribosome. In the specialised case of an mRNA truncated within the coding region, ribosomes stall at the 3' end with an empty A site that cannot be engaged by either elongation or termination complexes. This allows efficient engagement by the Pelota-Hbs1 complex to initiate a ribosome splitting reaction that separates the peptidyl-tRNA-60S complex from the mRNA-40S complex (*Becker et al., 2011*; *Pisareva et al., 2011*; *Shao et al., 2016*; *Shoemaker et al., 2010*). The 60S complex is engaged by ribosome-associated quality control (RQC) factors to trigger polypeptide degradation (*Shao et al., 2015*), while the 40S complex engages mRNA decay pathways by still poorly-understood mechanisms (*Schmidt et al., 2016*).

When a ribosome slows excessively within the coding region, its detection relies on ZNF598 (Hel2 in yeast), an E3 ubiquitin ligase whose ubiquitination of a 40S ribosomal protein (eS10 in mammals or uS10 in yeast) initiates downstream mRNA quality control (*Ikeuchi et al., 2019*) and protein quality control (*Garzia et al., 2017*; *Juszkiewicz and Hegde, 2017*; *Matsuo et al., 2017*; *Sundaramoorthy et al., 2017*). The cue used by ZNF598 to recognise a slow ribosome is the collision that occurs when a trailing ribosome catches up (*Juszkiewicz et al., 2018*). The collided ribosome shows a distinctive 40S-40S interface where ZNF598/Hel2 is proposed to bind and ubiquitinate nearby target sites (*Ikeuchi et al., 2019*; *Juszkiewicz et al., 2018*).

The steps downstream of ZNF598 are incompletely understood and appear to involve nuclease (*D'Orazio et al., 2019*), a helicase complex (*Hashimoto et al., 2020*; *Matsuo et al., 2017*; *Sitron et al., 2017*), and possibly ribosome rescue factors (*Tsuboi et al., 2012*). Although the roles and order of action of these factors is only partially resolved (*Juszkiewicz et al., 2020*; *Matsuo et al., 2020*), the requirement for ZNF598-mediated 40S ubiquitination (*Garzia et al., 2017*; *Juszkiewicz and Hegde, 2017*; *Matsuo et al., 2017*; *Sundaramoorthy et al., 2017*) together with the specificity of ZNF598 for ribosome collisions (*Juszkiewicz et al., 2018*) strongly argues that the collided ribosome is a key proxy of aberrant translation. Hence, collisions are an upstream event in both mRNA decay (*Simms et al., 2017*) and nascent protein quality control (*Ikeuchi et al., 2019*; *Juszkiewicz et al., 2018*).

Nearly all insight into ribosome stalling and the downstream consequences comes from the analysis of model substrates where specific stalling sequences have been introduced. However, the set of substrates that ordinarily engage RQC are poorly understood. The analysis of RNAs that crosslink to ZNF598 identified mRNAs, tRNAs, and rRNAs, consistent with its engagement of translating ribosome (*Garzia et al., 2017*). Strikingly, the tRNA that was strongly enriched decodes the AAA codon while no specific mRNAs were enriched above others. This observation can be rationalised because ribosomes are well established to stall during translation of poly(A) sequences (*Chandrasekaran et al., 2019*) and polyadenylation is known to occur inappropriately at near-cognate sites within the coding region (*Guydosh and Green, 2017*; *Ozsolak et al., 2010*; *Pelechano et al., 2013*). Thus, prematurely polyadenylated mRNAs are likely to be one physiologically relevant source of ribosome stalling, collisions, and ZNF598 engagement.

While the quality control events downstream of collisions have evolved to eliminate aberrant proteins and mRNAs, promiscuous triggering of these reactions from normal translation complexes would be detrimental. This might be why reticulocytes, which have exceptionally high ribosome densities on highly translated haemoglobin mRNAs, downregulate the collision sensor ZNF598 (*Juszkiewicz and Hegde, 2017*; *Mills et al., 2016*). Indeed, a detectable proportion of ribosomes in native reticulocyte polysomes are collided as judged by their ability to engage exogenously added ZNF598 and resistance to separation by exogenously added nuclease (*Juszkiewicz and Hegde, 2017*; *Juszkiewicz et al., 2018*).

Unlike the terminally differentiating reticulocyte with a minimal transcriptome and limited lifespan, most cells cannot forgo translation quality control. How cells either minimise incidental collisions or avoid promiscuous activation of quality control from them is not clear. Although the precise frequency of incidental collisions is not well established, they are probably not insignificant considering the wide variation in elongation rates (*Boersma et al., 2019*; *Ingolia et al., 2011*; *Riba et al., 2019*; *Yan et al., 2016*), combined with closely spaced ribosomes on heavily translated mRNAs (*Palade, 1955*). Indeed, recent ribosome profiling experiments examining collided di-ribosomes suggest widespread collisions at numerous (potentially physiologic) pause sites throughout the transcriptome (*Arpat et al., 2019*; *Han et al., 2020*; *Zhao et al., 2019*). Here, we have taken an unbiased proteomic approach to identify cellular factors that selectively engage ribosome collisions. In addition to ZNF598, we describe a previously unappreciated response to ribosome collisions that complements the RQC pathway by dynamically adjusting translation initiation.

## Results

### Proteomic identification of collision-specific ribosome interacting factors

We used quantitative proteomics (*Rauniyar and Yates, 2014*) to identify ribosome-associating factors recruited selectively to collisions. The poly-ribosome fractions were isolated from cells that were unperturbed, elongation arrested with high-dose emetine, or treated with low-dose emetine to incur widespread collisions (*Figure 1A*, inset). Tandem mass tag (TMT) mass spectrometry of these polyribosome fractions identified 598 proteins (*Figure 1—source data 1*). As expected, core ribosomal proteins were seen to be tightly clustered at the origin (orange circles) providing an internal control for the comparisons (*Figure 1A*). Only a handful of the 598 detected proteins were enriched on collided polysomes relative to both untreated polysomes (x-axis) and elongation-arrested polysomes (y-axis).

ZNF598 was the most enriched collision-specific interaction partner consistent with prior biochemical experiments in vitro and in cells (*Juszkiewicz et al., 2018*). To determine whether any of the other enriched proteins might participate in managing collisions via a ZNF598-independent pathway, we analysed ZNF598 knockout (KO) cells. The proteomes of globally stalled (with high-dose emetine) versus globally collided (with low-dose emetine) polysomes showed that only EDF1 and GIGYF2 remained enriched on collided ribosomes (*Figure 1B*). Thus, out of almost 600 proteins detected in polysomes, only two were identified to be enriched on collided polysomes relative to both stalled and unperturbed polysomes in both wild type and ZNF598-KO cells. We therefore investigated these further.

Immunoblotting of size-fractionated cytosol showed that both EDF1 and GIGYF2 are primarily observed in non-ribosomal fractions (*Figure 1C*). Importantly, long exposures of the blots showed that an estimated 5–10% of both proteins co-fractionated with polysomes. When cells were pre-treated with low-dose emetine, the polysome-associated populations of EDF1 and GIGYF2 sharply increased to 60% and 75%, respectively. Notably, no appreciable population of either protein was enriched in the mono-ribosome fractions. Parallel analysis in ZNF598 KO cells showed very similar fractionation patterns: a minor polysome-associated population of EDF1 and GIGYF2 in untreated cells that increases markedly in cells containing extensive ribosome collisions (*Figure 1D*).

Similar results were seen in reticulocyte lysate (*Figure 1—figure supplement 1A*), which is naturally devoid of ZNF598 due to its downregulation during erythrocyte differentiation (*Juszkiewicz et al., 2018*; *Mills et al., 2016*). Here, collisions were incurred on endogenous haemoglobin mRNAs by allowing elongation in the presence of excess inactive eRF1$^{AAQ}$ to prevent termination at the stop codon (*Brown et al., 2015*; *Juszkiewicz et al., 2018*). The binding of GIGYF2 to collided ribosomes was not via mRNA because its digestion with nuclease did not displace GIGYF2 from collided ribosomes (*Figure 1—figure supplement 1B*). Thus, in two different systems using two different methods to generate ribosome collisions, EDF1 and GIGYF2 are collision-specific ribosome interactors. These results corroborate the unbiased mass spectrometry analysis and establish that GIGYF2 and EDF1 engage ribosome collisions in a ZNF598-independent manner.

### GIGYF2 is a collision-dependent cis-acting translational repressor

Earlier studies have shown that GIGYF2 can inhibit translation initiation by recruiting the alternative cap-binding protein 4EHP (also known as EIF4E2) (*Morita et al., 2012*; *Peter et al., 2017*). Recruitment of 4EHP to an mRNA allows it to outcompete the normal cap-binding protein eIF4E to prevent initiation (*Chapat et al., 2017*; *Cho et al., 2005*). The mRNA targets of GIGYF2 are not well established. Our finding that GIGYF2 is a collision-specific factor suggested the hypothesis that mRNAs containing collided ribosomes are translationally repressed via GIGYF2-dependent 4EHP recruitment. Consistent with this idea, 4EHP was enriched ~1.5 fold on collided ribosomes relative to either normal translating polysomes or non-collided stalled polysomes (*Figure 1—source data 1*).

To test this idea, we monitored translation output from a reporter mRNA containing a collision-inducing ribosome stalling sequence (*Figure 2A*). This reporter contains GFP and a stalling sequence [21 consecutive lysine-encoding AAA codons, abbreviated as $(K^{AAA})_{21}$] separated by the viral 2A sequence (*Juszkiewicz and Hegde, 2017*). The 2A sequence causes skipping of peptide bond formation with continued downstream elongation, thereby releasing GFP before the ribosome stalls.

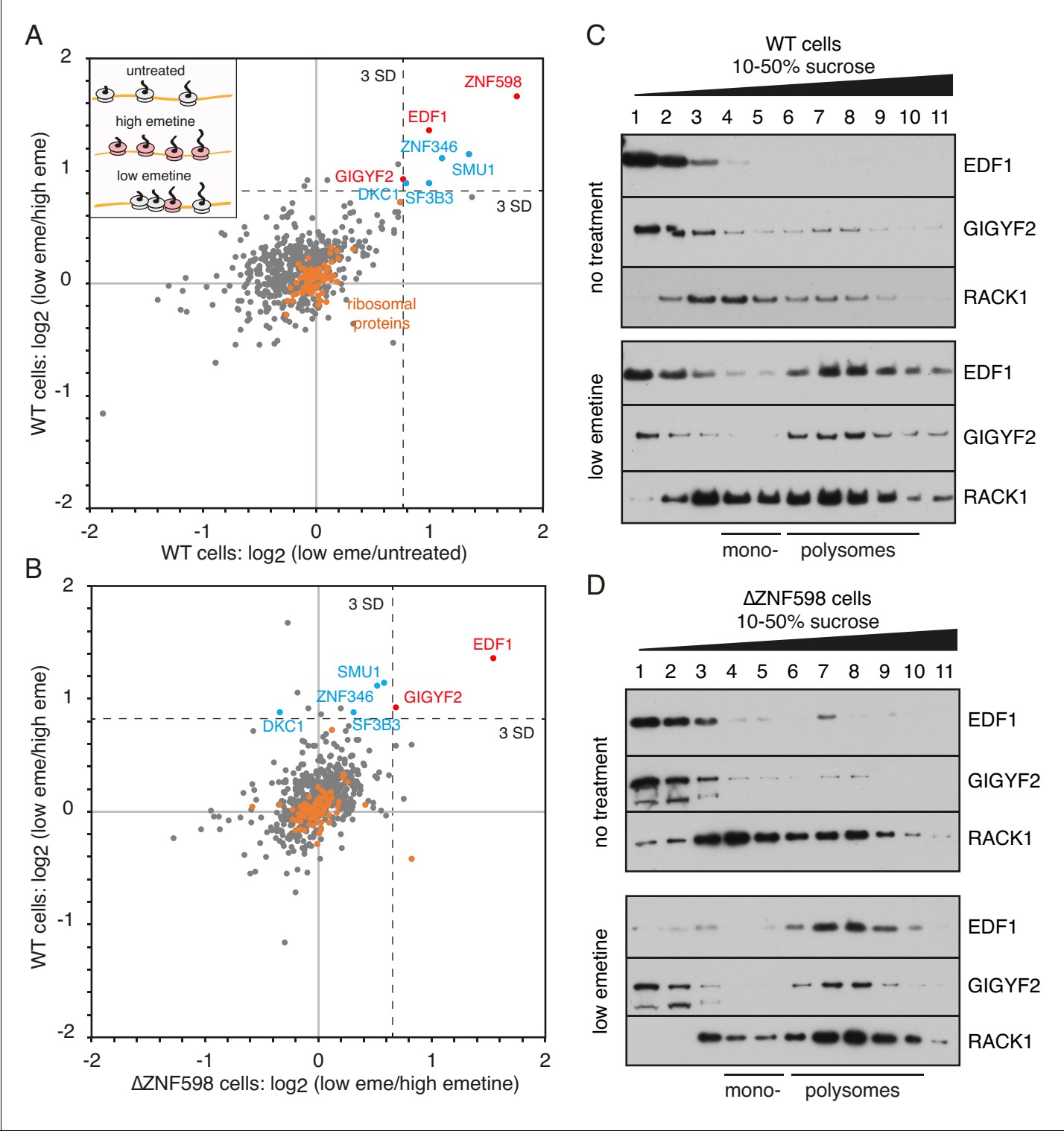

**Figure 1.** Proteomic identification of collision-specific ribosome interacting factors. (**A**) Plot of proteins identified by quantitative mass spectrometry in the polysome fraction of cytosol separated by sucrose gradient centrifugation. The cells were either left untreated, treated with low-dose emetine (1.8 µM) to induce collisions (low eme), or treated with high-dose emetine (360 µM) to freeze ribosomes in place (high eme). Pairwise comparisons are plotted such that factors specifically associating with collided ribosomes (i.e., only in low-dose emetine samples) should fall in the upper-right quadrant. Ribosomal proteins are indicated in orange. The position of 3 standard deviations (SD) from the mean along each axis is indicated with a dashed line. Proteins that deviate 3 SD or more along both axes are labelled (in blue and red). (**B**) Cells knocked out for ZNF598 (ΔZNF598) were treated with low or high dose emetine as in panel A and the polysome fractions were analysed by mass spectrometry. These data (x-axis) are plotted relative to the same

*Figure 1 continued on next page*

*Figure 1 continued*

comparison in wild type cells (y-axis). The position of 3 SD along each axis is indicated with a dashed line, and the data points for the same proteins from panel A are labelled. Note that only EDF1 and GIGYF2 fall into the upper right quadrant in both plots. As expected, ZNF598 was not detected in ΔZNF598 cells. (C, D) Sucrose gradient fractionation and immunoblots of cytosolic lysates prepared from untreated cells versus low-dose emetine treated cells. WT cells were analysed in panel C and ΔZNF598 cells in panel D. RACK1 is a 40S-ribosomal protein. The positions of fractions containing mono- and polysomes are indicated. Source data are provided for the mass spectrometry analysis.

The online version of this article includes the following source data and figure supplement(s) for figure 1:

**Source data 1.** Proteins identified by mass spectrometry of polysomes.

**Figure supplement 1.** Analysis of GIGYF2 and EDF1 interaction with collided ribosomes in rabbit reticulocyte lysate.

GFP levels therefore report on the number of translation events without being confounded by any quality control events triggered by stalling. The reporter also contains an RFP positioned after the stall sequence, allowing us to additionally monitor stall-induced disassembly versus read-through (which will be discussed later).

We found that GIGYF2 knockdown resulted in a ~2.5 fold increase in GFP output from the stalling reporter, but very little effect from a matched reporter lacking the stall sequence (*Figure 2B*). This effect could be ascribed to an effect on translation since quantitative RT-PCR showed no obvious effects on mRNA levels (*Figure 2—figure supplement 1A*). The small effect seen with the control reporter was consistently observed, which we suspected was due to a small degree of stalling (and ribosome collisions) at the slowly-decoded 2A sequence (*Sharma et al., 2012*). Consistent with this interpretation, a GFP-tagged protein lacking a 2A sequence showed no effect of GIGYF2 knock-down. Thus, the translation effect of GIGYF2 is substrate-specific and linked to ribosome stalling.

As with GIGYF2, knockdown of 4EHP also resulted in an increase in translation output selectively from the stall-containing reporter (*Figure 2C*). The more modest effect of 4EHP may be due to incomplete knockdown (*Figure 2—figure supplement 1B*). Regardless, the effect of 4EHP was dependent on GIGYF2 because no change in translation output was seen with 4EHP knockdown in GIGYF2 KO cells (*Figure 2D*). Consistent with GIGYF2 recruitment to collided ribosomes independently of ZNF598, the translational effects of GIGYF2 and 4EHP knockdown were observed in ZNF598 KO cells (*Figure 2E*). Considered together, these results indicate that GIGYF2 represses translation initiation via 4EHP consistent with earlier studies demonstrating their interaction (*Morita et al., 2012*). Because the effects are more prominent on stall-containing ribosomes that incur collisions, we further infer that GIGYF2 primarily operates in cis to inhibit translation from mRNAs to which it is recruited via collided ribosomes.

## Effect of EDF1 on frameshifting at stall sites

The other ZNF598-independent factor that we found to engage collided ribosomes is EDF1, a highly conserved factor found throughout the eukaryotic kingdom and in archaea. Deletion of the EDF1 homolog (Mbf1) in yeast was recently shown to cause increased frameshifting at ribosome stall sequences (*Wang et al., 2018*). Whether EDF1 has the same function in mammalian cells is not known. The stall sequences used in the yeast system, certain di-codon pairs or a stem-loop, do not detectably stall ribosomes in mammals (*Arthur et al., 2015*; *Juszkiewicz and Hegde, 2017*; *Sundaramoorthy et al., 2017*). Instead, mammalian ribosomes primarily stall on lengthy poly(A) sequences (*Arthur et al., 2015*; *Chandrasekaran et al., 2019*; *Garzia et al., 2017*; *Juszkiewicz and Hegde, 2017*; *Sundaramoorthy et al., 2017*). Unfortunately, poly(A) sequences that are sufficiently long to trigger a clear stall, such as $(K^{AAA})_{21}$, also cause extensive frameshifting (*Juszkiewicz and Hegde, 2017*). This precluded a clear test of EDF1's role using this stalling sequence.

We therefore tested a shorter poly(A) sequence [composed of $(K^{AAA})_{12}$] that causes some stalling, but less baseline frameshifting (*Juszkiewicz and Hegde, 2017*). In these three dual-colour GFP-RFP reporter constructs, the C-terminal RFP is appended directly after the poly(A) sequence in each reading frame (*Figure 3—figure supplement 1A*). As documented previously, such a trio of reporters lacking a stalling sequence shows near-undetectable levels of RFP when it is out of frame (*Juszkiewicz and Hegde, 2017*). By contrast, the $(K^{AAA})_{12}$ sequence inserted between GFP and RFP allows detectable RFP expression even for the out-of-frame constructs. Relative to the RFP signal for the in-frame construct, ~7% and ~13% signal was seen for the +1 and −1 frame constructs,

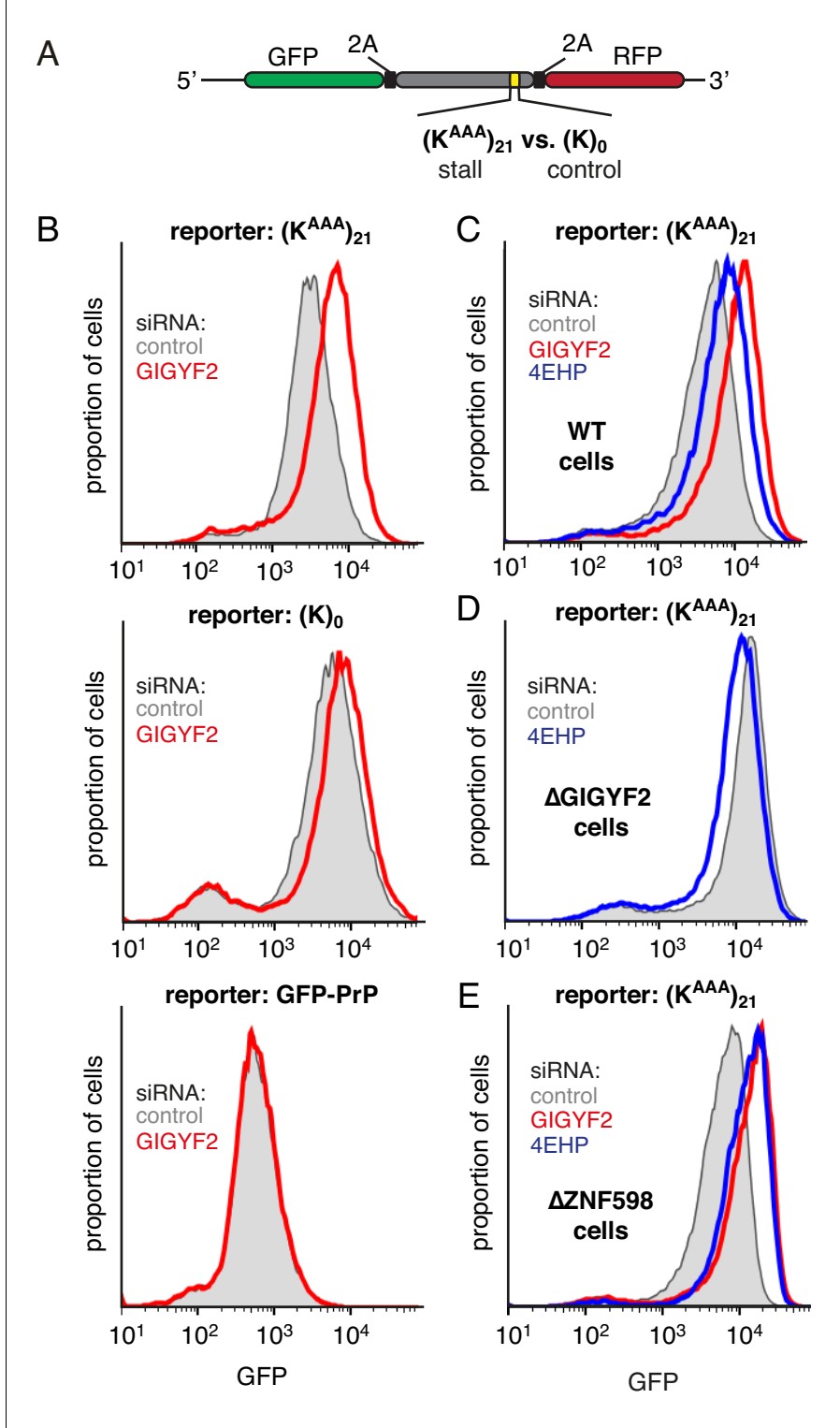

**Figure 2.** GIGYF2 is a collision-dependent cis-acting translational repressor. (**A**) Schematic representation of the reporter construct containing GFP followed by a viral 2A sequence and either the $(K^{AAA})_{21}$ stall sequence or a $(K)_0$ control insert that does not stall. (**B**) Flp-In T-REx 293 cells containing the indicated reporter construct stably integrated at a doxycycline-inducible site were treated for 5 days with either non-targeting control siRNAs (gray shaded) or siRNAs targeting GIGYF2 (red traces). The reporter construct was then induced with doxycycline for 20 hr prior to analysis by flow cytometry. GFP-PrP was used as an irrelevant fluorescent protein control to exclude non-specific effects on translation. (**C–E**) Flp-In T-REx 293 cells with stably intergrated $(K^{AAA})_{21}$ reporter were

*Figure 2 continued on next page*

*Figure 2 continued*

treated with siRNAs targeting GIGYF2 (red traces), 4EHP (blue traces) or control (gray shaded) as in panel A, induced with doxycycline for 20 hr, and analysed by flow cytometry. Panels D and E used cells that were knocked out for GIGYF2 or ZNF598, respectively. GFP expression in all graphs is plotted on a log scale as a histogram and represents around 20,000 cells.

The online version of this article includes the following figure supplement(s) for figure 2:

**Figure supplement 1.** Additional characterisation of knockdown and knockout cells.

respectively (*Figure 3—figure supplement 1B*). Upon siRNA-mediated knockdown of EDF1 (*Figure 3—figure supplement 1C*), little if any increase in the out-of-frame signal was observed (*Figure 3—figure supplement 1B*).

To determine whether a stronger stall could reveal a role for EDF1 in reading frame maintenance, we turned to stalling sequences derived from human XBP1 (*Yanagitani et al., 2011*) placed ahead of $(K^{AAA})_4$. A set of three reporters each were prepared for wild type XBP1, a stronger-stalling point mutant (S255A), and a weaker-stalling point mutant (W256A mutant) (*Figure 3A*). Using the in-frame versions, we confirmed the relative stalling potencies of these XBP1 sequences (*Figure 3B*) and verified that the stall-inducing variants are dependent on ZNF598 for aborting translation at the stall (*Figure 3C*). ZNF598-dependence indicates that ribosome collisions occur effectively at the stall-inducing sequences. Conversely, the ZNF598-independence of XBP1(W256A) indicates reduced or no collisions at this sequence, providing a negative control for frameshifting experiments.

Comparison of the three-frame set of non-stalling XBP1(W256A) reporters showed RFP signals of ~2% and ~9% in the +1 and −1 frames, respectively, relative to the in-frame construct (*Figure 3D*). This is consistent with some degree of frameshifting at the intrinsically 'slippery' $(K^{AAA})_4$ sequence, as expected from earlier studies (*Arthur et al., 2015*; *Koutmou et al., 2015*). Of note, this level of frameshifting is less than $(K^{AAA})_{12}$, as expected (compare to *Figure 3—figure supplement 1B*). Parallel analysis of the strong-stalling XBP1(S255A) constructs showed out-of-frame signals of 8% and 15% in the +1 and −1 frames, respectively (*Figure 3D*). This indicates that collisions enhance the extent of frameshifting. Similar results were seen for the medium-stalling XBP1 reporters (*Figure 3—figure supplement 2A*).

Upon knockdown of EDF1, the frameshifting frequency did not change appreciably for the non-stalling XBP1(W256A) reporter but clearly increased in the −1 frame for both the medium-stalling XBP1 and strong-stalling XBP1(S255A) reporters (*Figure 3D*, and *Figure 3—figure supplement 2A*). Notably, no obvious increase in the +1 frame was observed for these reporters in EDF1 knockdown cells. Similar results were seen for an independent siRNA that also depletes EDF1 effectively (*Figure 3—figure supplement 2B*). Of note, the GFP signal, indicative of mRNA levels and reporter translation, is very similar for the three different reading frame reporters within any experiment. Thus, nonsense-mediated decay, which in mammals typically requires an exon junction complex downstream of a premature stop codon (*Shoemaker and Green, 2012*), is not appreciably influencing our observations.

Our observations in mammalian cells roughly approximate recent observations in the yeast system: an increase in frameshifting at stalling sequences (*Simms et al., 2019*; *Wang et al., 2018*), which is enhanced by loss of Mbf1 (*Wang et al., 2018*). Nevertheless, the systems clearly differ. Whereas yeast show a dramatic ~300 fold increase in frameshifting without Mbf1 in some cases, the absence of EDF1 had rather modest consequences even downstream of a strong stalling sequence. Furthermore, the shifted frame that is preferred is also different: +1 in the case of di-codon triggered stalls in yeast versus −1 in the case of Xbp1-mediated stalling in mammalian cells. The basis of these differences may lie in the nature of the sequences used to cause stalling or differences between yeast and mammalian cells. The mode of stalling is likely to be at least part of the explanation because even within yeast, stalling caused by a stem-loop results in both −1 and +1 frameshifting, whereas stall-inducing di-codons result in +1 frameshifting. Additional work is needed to investigate the mechanistic basis of frameshifting at stalls in both yeast and mammals.

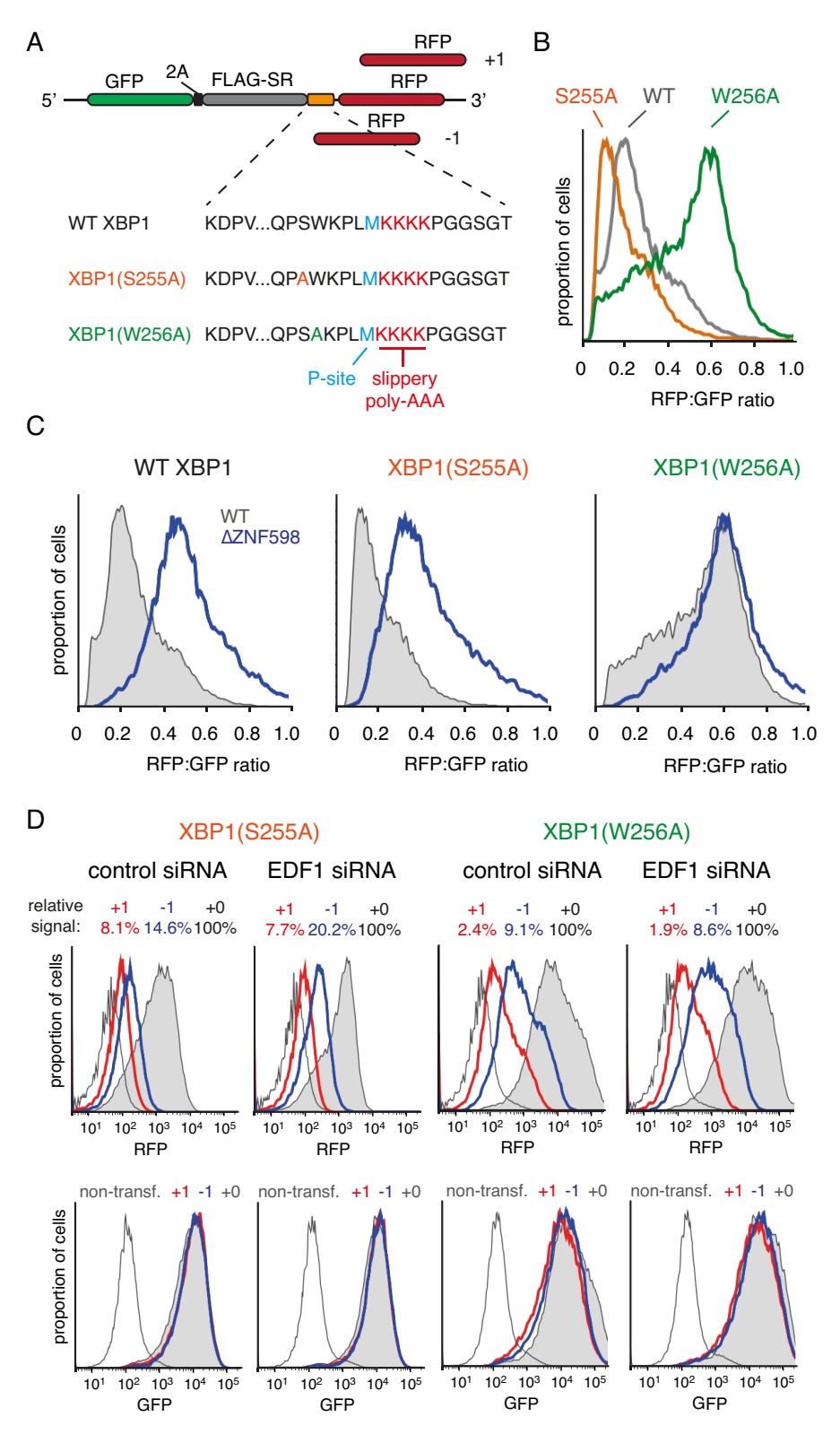

**Figure 3.** Effect of EDF1 on frameshifting at stall sites. (**A**) Schematic representation of the reporter constructs used to analyse frameshifting. RFP was directly appended to each of the three tested XBP1 reporters (see amino acid sequences below) in three different reading frames. Note that S255A mutation exacerbates, whereas W256A reduces ribosomal stalling, relative to the wild type XBP1 sequence. (**B**) In-frame versions of the constructs from panel A were co-transfected with BFP-encoding plasmid into Flp-In T-REx 293 cells and analysed by flow cytometry after 20 hr of transgene expression.

*Figure 3 continued on next page*

*Figure 3 continued*

At least 20,000 transfected (BFP-positive) cells were analysed here and in all the subsequent experiments involving frameshifting reporters. The RFP: GFP ratio, an indicator of readthrough at the XBP1 sequence, is plotted as a histogram for each reporter. (**C**) The same in-frame constructs as in panel B were analysed in WT and ZNF598 KO cells. Note that the RFP:GFP ratio is elevated in ΔZNF598 cells for WT XBP1 and XBP1(S255A), but not for XBP1 (W256A). (**D**) XBP1(S255A) (left) or XBP1(W256A) (right) was transfected into cells pre-treated for 72 hr with non-targeting siRNA (control) or EDF1-targeting siRNA and analysed by flow cytometry after 20 hr of reporter expression. Expression of GFP (bottom panels) and RFP (top panels) was plotted as histograms. Gray shaded traces represent cells positively transfected with in-frame (+0) constructs, blue traces represent −1 constructs and red traces represent +1 constructs. Background fluorescence observed in non-transfected cells is indicated by the unshaded thin gray traces. The amount of RFP signal observed for out-of-frame constructs (+1,−1), represented as a percentage of the in-frame signal, is indicated above each graph. Note that GFP expression is not influenced by the RFP reading frame.

The online version of this article includes the following figure supplement(s) for figure 3:

**Figure supplement 1.** Effect of EDF1 on frameshifting at stall sites.
**Figure supplement 2.** Effect of EDF1 on frameshifting at stall sites.

## EDF1 facilitates GIGYF2 function at collided ribosomes

While the physical interactions and consequences on frameshifting place EDF1's function at the ribosome, its molecular role there is unclear. Using our translational reporter assay, we observed that EDF1 knockdown modestly increased translation of a stall-containing mRNA, but had no effect on the control mRNA (*Figure 4A*). Increased translation with EDF1 knockdown was not ZNF598-dependent, but was completely lost in GIGYF2 KO cells. The combined knockdown of EDF1 and GIGYF2 was no greater than GIGYF2 alone (*Figure 4—figure supplement 1A*). These observations suggested that EDF1 has no effect on translation rates independently of GIGYF2; instead, EDF1 seems to facilitate GIGYF2 function.

Consistent with this idea, GIGYF2 interaction with polysomes was diminished in cells lacking EDF1 (*Figure 4B*). When EDF1 KO cells were treated with low-dose emetine to induce collisions, GIGYF2 was detectable but muted relative to WT cells. Reduced stability of GIGYF2 on collided ribosomes explains why loss of EDF1 partially phenocopies the GIGYF2 KO and why the combined loss is no different from loss of GIGYF2 alone.

Mutagenesis of Mbf1 in yeast has identified several point mutations impaired in its frameshifting suppression activity (*Wang et al., 2018*). To determine whether this phenotype might be related to its function at collided ribosomes, we tested whether the strongest of these mutants in EDF1 (R85G) influences ribosome binding. In vitro translated and affinity purified radiolabelled EDF1 was found to quantitatively engage collided ribosomes, but not monosomes, in reticulocyte lysate (*Figure 4C*). EDF1(R85G) showed substantially impaired interaction with collided ribosomes, with only ~40% co-fractionation.

WT EDF1, but not EDF1(R85G), rescued the stall-dependent translation phenotype in EDF1 KO cells (*Figure 4D*, *Figure 4—figure supplement 1B*). Thus, recruitment of EDF1 to collisions is important for its function in translational suppression via GIGYF2. Given the specificity of EDF1 interaction for collided ribosomes (*Figure 1*), the specificity of its frameshifting suppression on stalling sequences (*Figure 3*, *Wang et al., 2018*), and the correlation between collisions and frameshifting (*Simms et al., 2019*), it is attractive to posit that stall-induced collisions, frameshifting, and translation suppression are all mechanistically related.

In this view, collisions can increase the probability of frameshifting. EDF1 functions to reduce collision frequency by attenuating initiation frequency via GIGYF2-4EHP. Consistent with this idea, frameshifting at stall-induced collisions can be relieved by artificially suppressing the rate of translation initiation (*Simms et al., 2019*). Thus, the robust stall-induced frameshifting seen in yeast lacking Mbf1 may be a consequence of its exceptionally high translation rate during log-phase growth incurring a high rate of collisions. The role of Asc1 and Rps3 in suppressing frameshifting (*Wang et al., 2018*) might therefore be related to these proteins being needed for Mbf1 recruitment to collisions. Indeed, both proteins are located at the disome interface (*Ikeuchi et al., 2019*; *Juszkiewicz et al., 2018*). This working model warrants greater analysis.

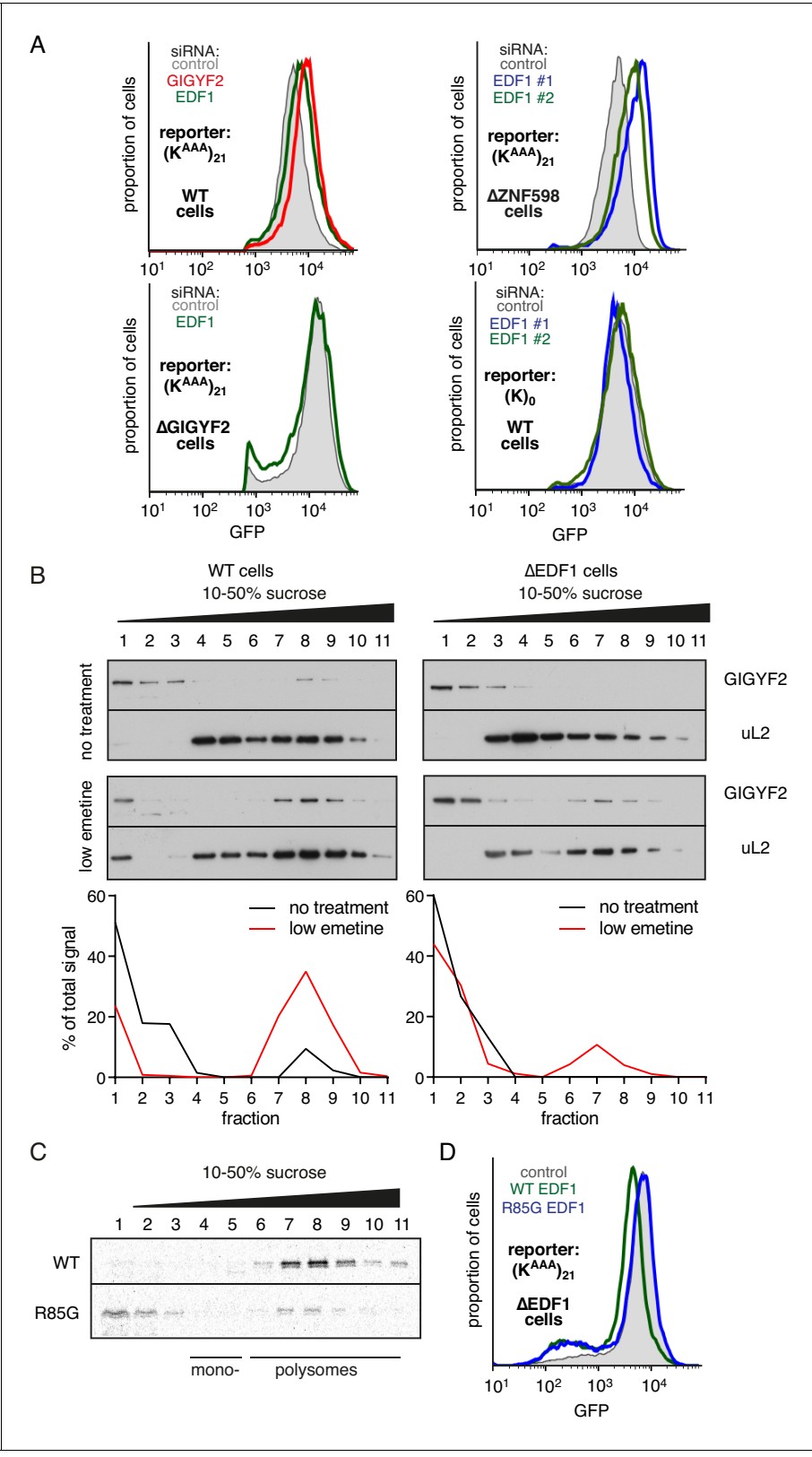

**Figure 4.** EDF1 facilitates GIGYF2 function at collided ribosomes. (**A**) The indicated cell lines bearing stably integrated translation reporters were treated with the indicated siRNAs for 72 hr. After additional 20 hr of doxycycline-induced reporter expression, cells were collected and analysed by flow cytometry. Histograms representing GFP expression were plotted on a logarithmic scale. (**B**) WT cells (left panel) and ΔEDF1 cells (right panel) were treated with low dose emetine (1.8 μM) for 1 hr and their cytosolic extracts were analysed by sucrose gradient centrifugation followed by western

*Figure 4 continued on next page*

*Figure 4 continued*

blotting. The profiles of GIGYF2 across the gradient, quantified by densitometry, are depicted below the blots. (**C**) Radiolabelled and immunopurified WT EDF1 or R85G mutant EDF1 produced by in vitro translation was added to separate non-labelled in vitro translation reactions in which ribosomes on native globin mRNA were stalled and collided at the stop codon with eRF1$^{AAQ}$. The samples were then separated by sucrose gradient centrifugation and EDF1 detected by autoradiography. Note that almost 100% of the WT EDF1 comigrates with collided ribosomes, whereas only ~40% of the R85G mutant is found in polysomal fractions with remainder at the top of the gradient. (**D**) ΔEDF1 cells with a stably integrated stalling reporter were transfected with a control plasmid (expressing BFP; shaded gray), WT EDF1 (green traces) and R85G EDF1(blue traces) and expression of transgenes was allowed for 24 hr. After additional 20 hr of doxycycline-induced expression of the stalling reporter, GFP expression was analysed by flow cytometry. Note that only WT, but not the R85G mutant reduces the expression of GFP to the level comparable to WT cells.

The online version of this article includes the following figure supplement(s) for figure 4:

**Figure supplement 1.** Additional characterisation of EDF1 function.

## ZNF598 and GIGYF2 mediate alternate responses to ribosome collisions

To understand the relationship between the collision sensor EDF1 and collision-specific effectors ZNF598 and GIGYF2, we first analysed their physical interactions using co-immunoprecipitation (IP). Earlier studies had shown interactions between ZNF598, GIGYF2, and 4EHP, with ZNF598 failing to co-IP with 4EHP when wild type GIGYF2 was replaced with a 4EHP-binding mutant (*Morita et al., 2012*). Because all of these proteins can interact with translation complexes, it was unclear whether the observed interactions reflected bridging via mRNA or ribosomes.

We therefore re-investigated potential interactions using cell lysates depleted of ribosomes. We found that FLAG-tagged ZNF598 could co-IP with GIGYF2 (*Figure 5A*). This interaction was unaffected by pre-treatment of cells with emetine (which, simply reduced the levels of GIGYF2 in the ribosome-free fraction). Notably, ZNF598 did not co-precipitate EDF1, and GFP-tagged EDF1 did not co-purify either ZNF598 or GIGYF2 (*Figure 5A*). Moreover, EDF1 was not necessary for the formation of ZNF598-GIGYF2 complex because the two proteins co-precipitated in EDF1 KO cells (*Figure 5B and C*). The lack of a ribosome-free interaction between EDF1 and GIGYF2 was verified with another tag (3xFLAG) (*Figure 5D*).

Thus, ZNF598 interacts with GIGYF2 independently of the ribosome, the latter of which directly interacts with 4EHP. ZNF598 apparently does not interact with 4EHP directly since the two proteins do not co-precipitate in the presence of a GIGYF2 mutant that cannot bind 4EHP (*Morita et al., 2012*). Although EDF1 is not part of this complex, it co-fractionates with and stabilises (but is not strictly required for) GIGYF2 on collided ribosomes (*Figure 4B*). This finding, together with the observation that EDF1 binding to collided ribosomes does not depend on either ZNF598 (*Figure 1*) or GIGYF2 (*Figure 5—figure supplement 1*), suggests a model where EDF1 engages collisions where it helps stabilise the binding of a pre-formed ZNF598-GIGYF2-4EHP complex (*Figure 5E*). Although we cannot know the order of binding to the ribosome from these data, we speculate that EDF1 binding may occur first because it is ~50 fold more abundant than either GIGYF2 or ZNF598 (*Itzhak et al., 2016*).

The two effectors that are recruited to ribosome collisions impact translation in different ways: GIGYF2 leads to inhibition of initiation, while ZNF598 leads to terminal stalling of elongation. To understand the functional relationship between these two outcomes, we analysed the role of GIGYF2 in ZNF598-mediated ribosome ubiquitination and ribosome stalling. Using cell lines knocked out for either GIGYF2 or EDF1 (*Figure 6—figure supplement 1*), we found that stall-triggered eS10 ubiquitination was largely unimpaired (*Figure 6A and B*), unless ZNF598 was also depleted (*Figure 6C*). Thus, ZNF598's ability to selectively recognise and mark collided ribosomes does not depend on EDF1 or GIGYF2.

This result might initially seem surprising in light of fractionation experiments that showed marked impairment of ZNF598 co-sedimentation with collided ribosomes in cells lacking EDF1 or GIGYF2 (*Figure 6—figure supplement 2*). On long exposures of the blots, collision-specific co-sedimentation could be seen. This is similar to the finding that GIGYF2 is strongly impaired in binding collisions in cells lacking EDF1 (*Figure 4B*). Yet, EDF1 depletion largely phenocopies GIGYF2 loss (*Figure 4—figure supplement 1*) while ZNF598 ubiquitination of eS10 is unimpaired (*Figure 6A, B and C*). This can be rationalised because ubiquitination by ZNF598 only requires a brief interaction (as is typical

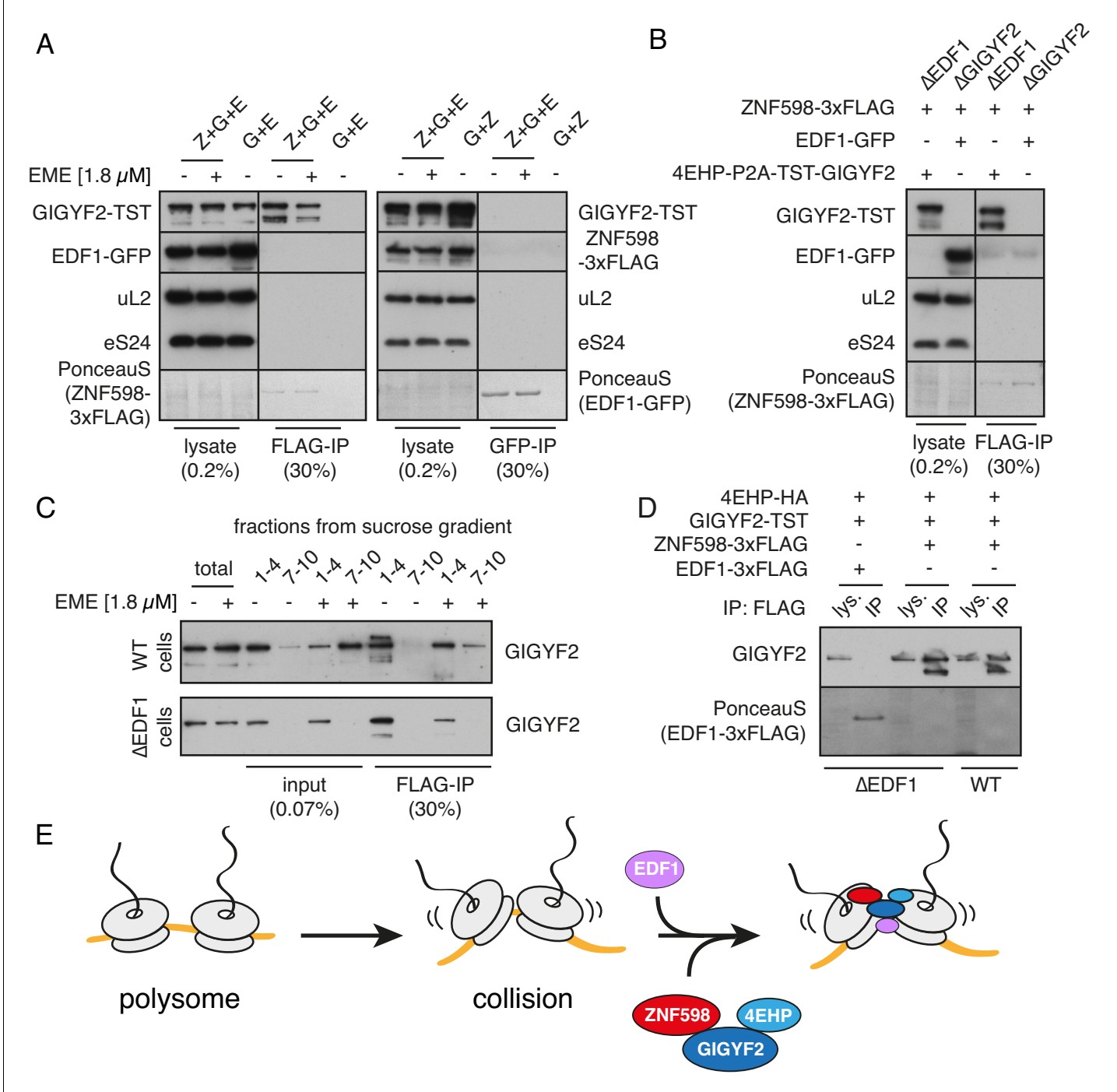

**Figure 5.** Analysis of interactions between EDF1, ZNF598, and GIGYF2. (**A**) WT HEK293 cells were transfected with constructs encoding indicated proteins (Z- ZNF598-3xFLAG; G - 4EHP-P2A-TST-GIGYF2; E - EDF1-GFP) and allowed to express for 20 hr. Cytosol was depleted of ribosomes by sedimentation and subjected to immunoprecipitation (IP) of 3xFLAG-ZNF598 (left panel) or EDF1-GFP (right panel). When indicated, transfected cells were additionally pre-treated with 1.8 μM emetine for 30 min prior to lysis. Total lysate as well as eluates after IP were analysed by western blotting. The stained blot verifies that the immunoprecipitated proteins were recovered. The IP lanes for each antigen (and the stained blot) are from the same gel and exposure as the lysate lanes, with intervening lanes removed digitally. The relative amounts of lysate and IP samples are indicated on the figure. (**B**) Cells lacking EDF1 (ΔEDF1) or GIGYF2 (ΔGIGYF2) were transfected with indicated plasmids and analysed as in panel A. (**C**) WT (top panel) or ΔEDF1 (bottom panel) cells were transfected with ZNF598-3xFLAG and 4EHP-P2A-TST-GIGYF2 encoding plasmids. After 20 hr of expression and, when indicated, pre-treatment with 1.8 μM emetine, cell lysates were fractionated on a sucrose gradient. Ribosome free fractions 1–4 and heavy polysome fractions 7–10 were pooled and each was analysed by FLAG-IP and western blotting. (**D**) WT or ΔEDF1 cells were transfected with indicated plasmids and IP was performed as in panel B. Note that EDF1 tagged with 3xFLAG does not recover GIGYF2 (lane 2), unlike ZNF598 containing the same tag

*Figure 5 continued on next page*

*Figure 5 continued*

(lanes 4 and 6). (**E**) ZNF598-GIGYF2-4EHP form a complex independent of the ribosome. This complex does not include or require EDF1. However, both EDF1 and the ZNF598-GIGYF2-4EHP engage collided ribosomes. Without EDF1, the ZNF598-GIGYF2-4EHP complex is less stably bound to collided ribosomes.

The online version of this article includes the following figure supplement(s) for figure 5:

**Figure supplement 1.** EDF1 recruitment to collided ribosomes in cells lacking GIGYF2.

for ubiquitin ligases and other enzymes) while effective competition for the 5' cap by GIGYF2-4EHP would require its constant presence at collisions. Thus, although EDF1 is needed for stabilisation of both ZNF598 and GIGYF2 at collided ribosomes (*Figure 5E*), EDF1 loss preferentially impacts the output from GIGYF2.

Functional analysis of ZNF598 used the dual colour stalling reporter combined with siRNA knockdowns (*Figure 6D*). As expected from earlier studies, ZNF598 knockdown results in increased readthrough of the $(K^{AAA})_{21}$ stall to generate higher levels of RFP. The slight decrease in GFP, which had previously escaped our attention, might reflect slightly improved GIGYF2 function on collisions lacking ZNF598. With GIGYF2 knockdown, GFP levels increased as expected, but almost no increase in RFP was seen. This indicates a decreased frequency of readthrough, suggesting enhanced ZNF598 function on collisions lacking GIGYF2. The double knockdown showed a combination of the two phenotypes: increased GFP indicating greater translation initiation, and increased RFP indicating greater read-through. Thus, neither ZNF598 nor GIGYF2 depend on the other for function, but each factor's function is slightly enhanced when the other is absent.

Overexpression studies complemented the knockdown experiments and supported these conclusions (*Figure 6—figure supplement 3*). Overexpression of GIGYF2 together with its binding partner 4EHP resulted in decreased GFP expression indicative of reduced initiation and increased RFP indicative of greater readthrough. The RFP:GFP ratio was therefore substantially elevated, arguing that ZNF598 function in stalling translation was impaired. Conversely, ZNF598 overexpression resulted in increased GFP consistent with impaired GIGYF2 function. While RFP was also increased, the overall RFP:GFP ratio was reduced consistent with enhanced ZNF598 activity. In cells knocked out for either ZNF598 or GIGYF2, overexpression of the other had no appreciable effects. This argues that in these cells, the remaining factor is functioning at maximal capacity so the introduction of additional factor has no consequence. Thus, in wild type cells, ZNF598 partially antagonises the effect of GIGYF2 and vice versa. Inspection of the RFP, GFP, and ratio plots of the flow cytometry experiments from *Figure 4* (*Figure 6—figure supplement 4*) support this conclusion.

## Discussion

Cells exploit ribosome collision as an indirect indicator of an aberrant mRNA that cannot be successfully translated (*Ikeuchi et al., 2019*; *Juszkiewicz et al., 2018*; *Simms et al., 2017*). Accordingly, the recruitment of ZNF598 to such collisions can initiate mRNA and protein quality control pathways. However, collisions can also indicate the more innocuous situation of high ribosome density caused by overly frequent initiation. Our discovery of a collision-dependent cis-acting mechanism of inhibiting translation initiation suggests that cells monitor ribosome density at a per-mRNA level and dynamically tune initiation rates accordingly. Thus, ribosome collisions emerge as an important parameter that cells use to adjust the quality and quantity of translational output.

At an average density of one ribosome per ~60 codons and a highly variable translation rate that averages ~6 codons per second, collisions are inevitable (*Reuveni et al., 2011*; *Tuller et al., 2010*). This is because considerations of optimal resource utilisation dictate that translation efficiency and ribosome density should be at slightly sub-saturation levels (*Reuveni et al., 2011*). Consistent with this, electron microscopy images have long shown polysomes contain ribosomes that are typically separated by ~8–15 nm (*Palade, 1955*). In HEK293 cells growing under laboratory conditions, we estimate that ~2–5% of ribosomes are collided at any given moment based on the small but discernible disome peak seen on sucrose gradients after nuclease digestion (*Juszkiewicz et al., 2018*). These considerations suggest that a general detector of collisions must be within ~20 fold abundance of ribosomes.

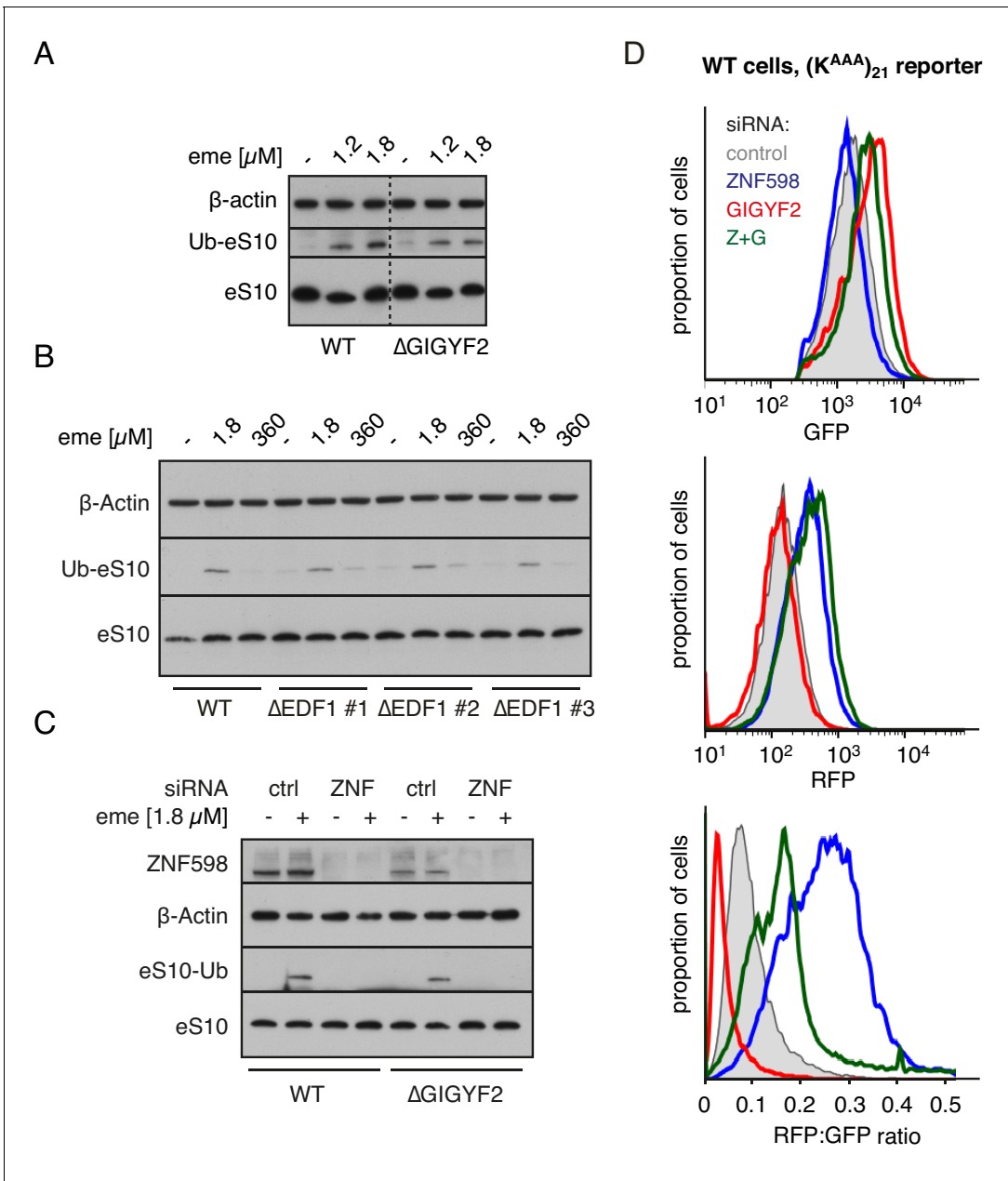

**Figure 6.** ZNF598 and GIGYF2 mediate alternate responses to ribosome collisions. (A) Total cell lysates from WT and ΔGIGYF2 cells treated with nothing or with collision-inducing low-dose emetine (1.2 μM or 1.8 μM) were analysed by western blotting to monitor ubiquitination of eS10, the primary target of ZNF598. Note that due to its low abundance, the strip of the blot containing ubiquitinated eS10 (Ub-eS10) was detected with higher concentration of primary antibody, but is otherwise from the same membrane and gel used for the other antigens. (B) WT and ΔEDF1 cells were treated with nothing, collision-inducing low dose of emetine (1.8 μM) or stall inducing high dose of emetine (360 μM). The lysates were analysed by western blotting to assess ubiquitination of eS10 as in panel A. (C) WT and ΔGIGYF2 cells were treated with indicated siRNAs for 72 hr. Where indicated, cells were treated with 1.8 μM emetine for 15 min just prior to lysis. Shown are total cell lysates analysed by western blotting to asses ubiquitination of eS10. Note that in both cell types, eS10 ubiquitination is completely abolished when ZNF598 is knocked down. (D) Flp-In T-REx 293 with stably integrated $(K^{AAA})_{21}$ reporter were treated with the indicated siRNAs for 3 days. After an additional 20 hr of doxycycline-induced expression of the fluorescent reporter, the cells were analysed by flow cytometry. Expression of GFP (top panel) and RFP (middle panel) was represented as a histogram for each condition plotted on a log scale. The RFP:GFP ratio (bottom panel) was calculated for each cell line using FlowJo software and also represented as a histogram for each condition.

The online version of this article includes the following figure supplement(s) for figure 6:

**Figure supplement 1.** Validation of EDF1 and GIGYF2 knockout cell lines.

**Figure supplement 2.** ZNF598 recruitment to collided ribosomes in different cell lines.

*Figure 6 continued on next page*

*Figure 6 continued*

**Figure supplement 3.** Effect of ZNF598 and GIGYF2 overexpression.
**Figure supplement 4.** Additional analysis of dual-colour stalling reporters.

Quantification in Hela cells indicates that EDF1 is present in ~1 million copies per cell (*Kulak et al., 2014*), roughly one-fourth to one-sixth the concentration of ribosomes as measured in the same study. By contrast, GIGYF2 and ZNF598 are estimated to be around 50-fold lower (*Itzhak et al., 2016*), although this may vary by cell type. We therefore posit that EDF1 is a general collision sensor that arrives first at any collision due to its very high abundance. If the collision is simply due to stochastic variations in ribosome spacing along an mRNA, elongation would continue, the ribosome spacing would quickly change, and EDF1 would dissociate without any action being taken. Because each round of elongation only takes ~200 milliseconds [i.e., an average of ~5 amino acids per second (*Sokabe and Fraser, 2019*)], these events would all be complete within a second or so.

If the collision is due to a genuine slowdown, EDF1 would be retained on collided ribosomes for longer, thereby stabilising the ZNF598-GIGYF2-4EHP complex at the collision (*Figure 7*). While GIGYF2 co-fractionation with collided ribosomes is diminished without EDF1, it is not lost entirely. Similarly, ZNF598 binding to collided ribosomes is substantially less stable but not eliminated in the absence of either EDF1 or GIGYF2. In each case, the ribosome interaction is collision-specific. This is consistent with a model in which the disome-EDF1 complex provides an optimal platform for ZNF598-GIGYF2-4EHP binding. This could occur in one of two ways: (i) a direct model where EDF1 provides part of a binding site for the ZNF598-GIGYF2-4EHP complex on collided ribosomes; (ii) an indirect model where EDF1 engagement of collided ribosomes changes the ribosome complex in a way that favours the bound state of the ZNF598-GIGYF2-4EHP complex. Future structural analysis of the complex engaged on a collided disome is required to address these details.

Once recruited, GIGYF2 can repress translation initiation. As established previously, GIGYF2 forms a complex with 4EHP (*Morita et al., 2012*), which competes with the translation factor eIF4E for the 5' cap (*Cho et al., 2005*). 4EHP is ordinarily not able to outcompete eIF4E (*Zuberek et al., 2007*) unless its local concentration in the vicinity of an mRNA is high. Such a model would explain how collisions can selectively inhibit translation initiation of the mRNA on which the collision occurred. Consistent with this idea, artificially recruiting the GIGYF2-4EHP complex to an mRNA is sufficient to inhibit translation initiation in cis (*Hickey et al., 2019*).

Because incidental collisions may be a pervasive feature across the transcriptome (*Arpat et al., 2019*; *Han et al., 2020*), the GIGYF2-4EHP axis is expected to impact the translation of a wide range of mRNAs. Earlier experiments in cells lacking GIGYF2 or 4EHP showed ~30% higher rates of translation globally (*Morita et al., 2012*). Strikingly, the native size of polysomes in 4EHP knockout tissue was increased by the size of 1–2 ribosomes, indicating that ribosome density on mRNAs is increased across the transcriptome.

Reduction of initiation rates via the GIGYF2-4EHP axis provides time for resolution of the collision without incurring further collisions. Resolution can occur in one of two ways. If the collision is incidental due to a physiologic pause in translation, elongation would soon resume, at which point EDF1 and the ZNF598-GIGYF2-4EHP complex would dissociate to allow initiation to begin again (*Figure 7*, left). Given the abundance and speed of elongation factor action, resumption of translation would typically occur in under a second. The broadest estimates of normal elongation rates span at most a ~20 fold range (*Rodnina, 2016*; *Sokabe and Fraser, 2019*). Thus, even in particularly unusual instances where a ribosome lingered for 20-fold longer than a normal elongation cycle, it would still re-start in ~4 s.

The longer a collision persists, the greater the likelihood that ZNF598 will ubiquitinate 40S proteins including eS10 (*Figure 7*, right). Depending on the ligase and other parameters, ubiquitination reactions can occur on the timescale of milliseconds to minutes (*Pierce et al., 2009*; *Melvin et al., 2013*). A time course of ZNF598 ubiquitination of ribosomes in vitro at close to physiologic concentrations indicate that it takes at least ~2–5 min to accumulate appreciable 40S ubiquitination (*Juszkiewicz and Hegde, 2017*). Because there are likely to be deubiquitinase(s) that can act on the ribosome (*Garshott et al., 2020*), 40S ubiquitination is not the commitment step to trigger RQC.

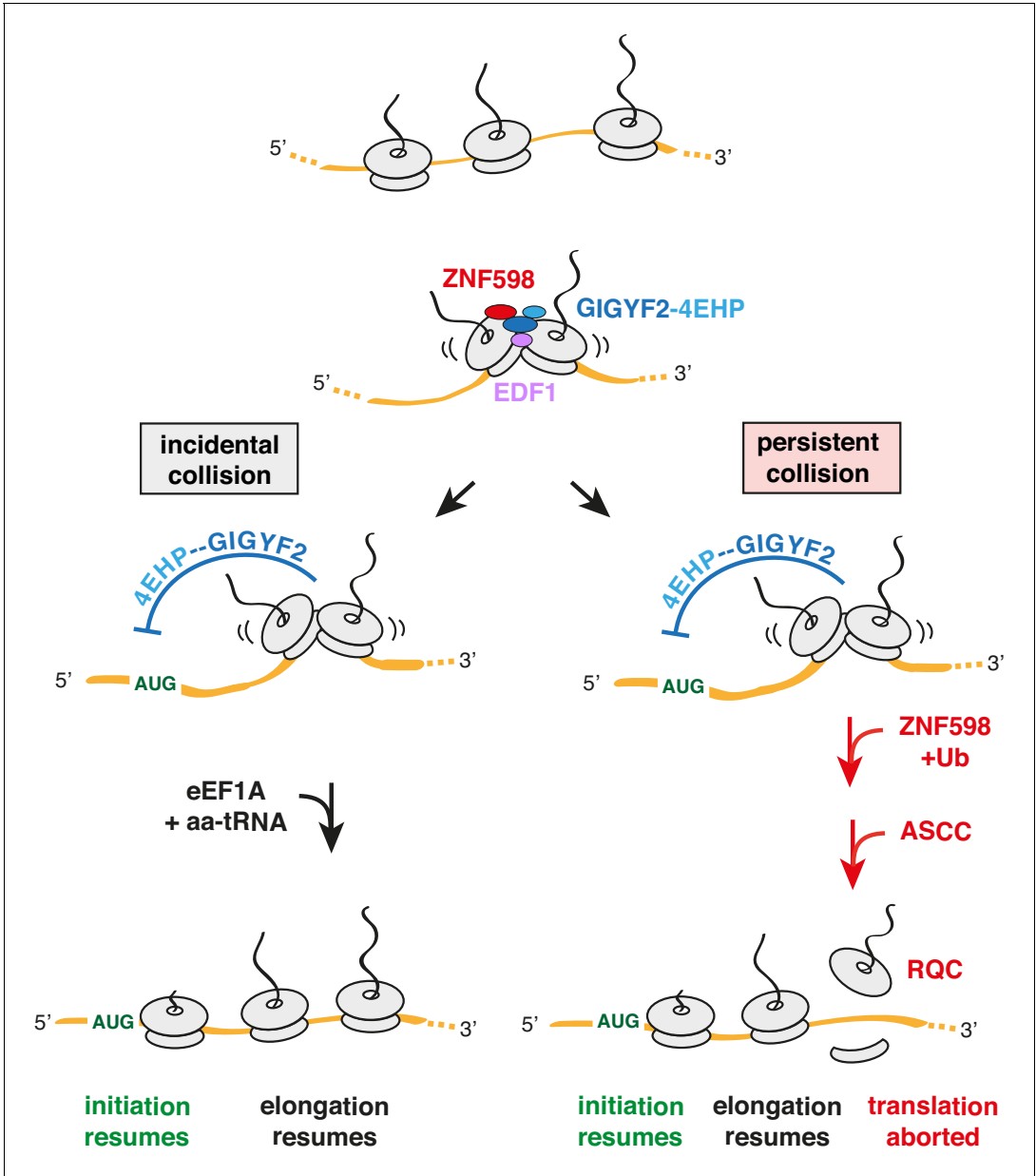

**Figure 7.** Model for tiered response to ribosome collisions. When two ribosomes collide, EDF1 is most likely to engage the collision first due to its high abundance. Because downstream responses require recruitment of additional factors, no action is taken if the collision resolves quickly by elongation of the lead ribosome. This allows for random bumping of ribosomes on a crowded message. However, if resolution is not immediate, EDF1 stabilises the ZNF598-GIGYF2-4EHP complex. 4EHP inhibits initiation via sequestration of the 5' cap to avoid further increase in ribosome density. This inhibition is relieved in one of two ways. If the collision is incidental (left limb), as might occur at sites of physiologic ribosome pausing, elongation resumes through the action of eEF1A in complex with aminoacyl-tRNA (aa-tRNA). If the collision is persistent (right limb), as would occur at sites of pathologic stalling, ZNF598 ubiquitinates the ribosome, which signals the ASC-1 complex (ASCC) to disassemble the lead ribosome. The 60S complex of the disassembled lead ribosome engages the ribosome quality control (RQC) complex. Because the stalled ribosome and ensuing collision has been resolved, elongation of trailing ribosomes resumes, causing dissociation of EDF1 and the ZNF598-GIGYF2-4EHP complex. Loss of 4EHP permits initiation to resume. Ub is ubiquitin.

Instead, ubiquitinated eS10 is needed for the ASC-1 complex (ASCC) to act (*Juszkiewicz et al., 2020*; *Matsuo et al., 2020*).

ASCC is a conserved complex containing the ASCC3 helicase (Slh1 in yeast) involved in resolving collided ribosomes (*D'Orazio et al., 2019*; *Hashimoto et al., 2020*; *Ikeuchi et al., 2019*; *Juszkiewicz et al., 2020*; *Matsuo et al., 2017*; *Matsuo et al., 2020*). Recent experiments suggest

that by directly dissociating the lead ribosome into subunits, ASCC is the factor that irreversibly aborts translation and initiates RQC on the stalled ribosome (*Juszkiewicz et al., 2020*; *Matsuo et al., 2020*). Having resolved the troublesome ribosome, the downstream ribosomes are then free to elongate (*Juszkiewicz et al., 2020*).

Because the commitment to RQC is two steps removed from ZNF598 recruitment and involves the trans-acting ASCC, this limb of the response is likely to take between one to many minutes. By contrast, the simple binding reaction represented by 4EHP interacting with the 5' cap in cis would occur at millisecond to low-second time scales given the constrained local concentrations of the interaction partners. Although we have not made all of these measurements, their estimates are based on well-established biochemical principles. Thus, we propose that collisions initiate a tiered response that begins with the conservative strategy of inhibiting initiation (the 1° response) followed later by the destructive strategy of initiating quality control (the 2° response). If the lead ribosome elongates at any point prior to ASCC action, the distinctive architecture of the collided disome would be lost, causing dissociation of the EDF1 and the ZNF598-GIGYF2-4EHP complex.

Even though ZNF598 is more stable on collisions containing EDF1 and GIGYF2, ZNF598-mediated eS10 ubiquitination does not require either protein. Hence, RQC can be effectively triggered without EDF1 or GIGYF2. Similarly, GIGYF2 does not require ZNF598 for either its stability on collisions or its effect on translation initiation. While future structural studies are needed to understand the physical relationship of these factors on the collided ribosome, our biochemical and functional analyses suggest that they do not compete for a common site.

Nevertheless, GIGYF2 and ZNF598 do functionally compete for the outcome downstream of ribosome collisions. This can be rationalised because each pathway reduces collisions needed by the other pathway's output. The ZNF598 limb reduces collisions by ultimately splitting the lead ribosome. Thus, increased output through this limb means fewer collided ribosomes from which the GIGYF2-4EHP complex can inhibit initiation. On the other hand, increased output through the GIGYF2-4EHP limb reduces the initiation rate, reducing collisions on which ZNF598 can act.

The implication of this tiered model is that the ratio of GIGYF2 to ZNF598 effectively sets the persistence threshold at which collisions trigger quality control. This key ratio may vary across cell types as evidenced by the lack of ZNF598 in reticulocytes (*Juszkiewicz et al., 2018*; *Mills et al., 2016*) and its particularly high expression in oocytes (*Grennan, 2006*; www.genevisible.com). Reticulocytes may forego quality control in order to translate at maximal levels, while other cell types might show particularly sensitive quality control to minimise aberrant proteins. The regulatory aspect of how cells respond to collisions remains to be explored.

Although relatively little was previously known about EDF1, the yeast homolog Mbf1 was identified in a screen for genes that suppress frameshifting at ribosome stalling sequences (*Wang et al., 2018*). Given the recent finding that suggests ribosome collisions can cause frameshifting (*Simms et al., 2019*), our results provide a plausible explanation for the increased frameshifting observed in ΔMbf1 yeast strains. By removing a factor that helps to suppress initiation in response to collisions, the collision rate would increase, resulting in increased frameshifting. Consistent with this idea, yeast has two GIGYF2 homologs (Smy2 and Syh1) whose overexpression suppresses translation of a stall-containing reporter (*D'Orazio et al., 2019*). Although yeast does not have an obvious 4EHP homolog, Smy2 interacts with Eap1, an inhibitor of eIF4E function (*Sezen et al., 2009*). Thus, it is likely that the cis-acting collision-mediated feedback loop described in our study is widely conserved.

If each collision is accompanied by a slight increase in the probability of frameshifting, minimising their occurrence would be important for avoiding the production of aberrant proteins. Whether the accumulation of aberrant frameshifted proteins is the basis for the neurodegenerative-like phenotype of partial GIGYF2 deficiency (*Giovannone et al., 2009*) remains to be explored. It is particularly intriguing that this phenotype resembles in some ways the partial deficiency of Listerin (*Chu et al., 2009*), which acts downstream of ZNF598 (*Garzia et al., 2017*; *Juszkiewicz and Hegde, 2017*; *Sundaramoorthy et al., 2017*). Thus, by acting to minimise (via the GIGYF2 limb) and resolve (via the ZNF598 limb) ribosome collisions, both pathways seem to protect the proteome from detrimental translation products that may contribute to neurodegenerative disease.

# Materials and methods

## Key resources table

| Reagent type (species) or resource | Designation | Source or reference | Identifiers | Additional information |
|---|---|---|---|---|
| Cell line (*H. sapiens*) | HEK293T | ATCC | CRL-3216; RRID:CVCL_0063 | |
| Cell line (*H. sapiens*) | Flp-In T-REx 293 | Thermo Fisher | R78007; RRID:CVCL_U427 | |
| Cell line (*H. sapiens*) | Flp-In T-REx 293 GFP-P2A-(KAAA)21-P2A-RFP | *Juszkiewicz and Hegde, 2017* | | |
| Cell line (*H. sapiens*) | Flp-In T-REx 293 TRex GFP-P2A-(K)0-P2A-RFP | *Juszkiewicz and Hegde, 2017* | | |
| Cell line (*H. sapiens*) | Flp-In T-REx 293 GFP-P2A-(KAAA)21-P2A-RFP ΔZNF598 | *Juszkiewicz and Hegde, 2017* | | |
| Cell line (*H. sapiens*) | Flp-In T-REx 293 GFP-P2A-(KAAA)21-P2A-RFP ΔGIGYF2 | This paper | | CRISPR/Cas9 targeting GIGYF2, clonal selection |
| Cell line (*H. sapiens*) | Flp-In T-REx 293 GFP-P2A-(KAAA)21-P2A-RFP ΔEDF1 | This paper | | CRISPR/Cas9 targeting EDF1, clonal selection |
| Antibody | anti-ZNF598 (rabbit polyclonal) | Abcam | Cat #ab80458; RRID:AB_2221273 | WB (1:500) |
| Antibody | anti-uL2 (rabbit polyclonal) | Abcam | Cat #ab169538; RRID:AB_2714187 | WB (1:10000) |
| Antibody | anti-eS24 (rabbit polyclonal) | Abcam | Cat #ab196652; RRID:AB_2714188 | WB (1:2500) |
| Antibody | HRP-conjugated anti-FLAG (mouse monoclonal) | Sigma | cat #A8592; RRID:AB_439702 | WB (1:5000) |
| Antibody | anti-eS10 (rabbit monoclonal) | Abcam | cat #ab151550; RRID:AB_2714147 | WB (1:5000 – unmodified eS10; 1:250 – Ub-eS10) |
| Antibody | HRP-conjugated anti-beta-Actin (mouse monoclonal) | Sigma | cat #A3854; RRID:AB_262011 | WB (1:10000) |
| Antibody | anti-GIGYF2 (rabbit polyclonal) | Bethyl | cat #A303-732A; RRID:AB_11205815 | WB (1:2000) |
| Antibody | anti-EDF1 (rabbit polyclonal) | Bethyl | cat #A304-039A; RRID:AB_2621288 | WB (1:1000) |
| Antibody | anti-RACK1 (rabbit polyclonal) | Bethyl | cat #A302-545A; RRID:AB_1999012 | WB (1:2000) |
| Antibody | anti-4EHP (rabbit polyclonal) | GeneTex | cat #GTX103977; RRID:AB_2036842 | WB (1:250) |
| Antibody | anti-GFP (rabbit polyclonal) | *Chakrabarti and Hegde, 2009* | | WB (1:5000) |
| Antibody | HRP-conjugated anti-mouse (goat polyclonal) | Jackson Immuno Research | cat #115-035-003; RRID:AB_10015289 | WB (1:5000) |
| Antibody | HRP-conjugated anti-rabbit (goat polyclonal) | Jackson Immuno Research | cat# 111-035-003; RRID:AB_2313567 | WB (1:5000) |
| Recombinant DNA reagent | pX459-EDF1-sgRNA1 | This paper | | sgRNA targeting EDF1 |
| Recombinant DNA reagent | pX459-EDF1-sgRNA2 | This paper | | sgRNA targeting EDF1 |
| Recombinant DNA reagent | pX459-GIGYF2- sgRNA1 | This paper | | sgRNA targeting GIGYF2 |

*Continued on next page*

*Continued*

| Reagent type (species) or resource | Designation | Source or reference | Identifiers | Additional information |
|---|---|---|---|---|
| Recombinant DNA reagent | pX459-GIGYF2- sgRNA2 | This paper | | sgRNA targeting GIGYF2 |
| Recombinant DNA reagent | pcDNA3.1-ZNF598-TEV-3xFLAG | *Juszkiewicz and Hegde, 2017* | RRID:Addgene_105690 | Human ZNF598 with TEV-3xFLAG tag at the C-terminus for mammalian expression |
| Recombinant DNA reagent | pcDNA3.1-EDF1-TEV-3xFLAG | This paper | | Human EDF1 tagged with TEV-3xFLAG at the C-terminus for mammalian expression |
| Recombinant DNA reagent | pcDNA3.1-EDF1-GFP | This paper | | Human EDF1 tagged with GFP at the C-terminus for mammalian expression |
| Recombinant DNA reagent | pCMV-4EHP-HA | This paper | | Human 4EHP tagged with HA at the C-terminus for mammalian expression |
| Recombinant DNA reagent | pcDNA3.1-4EHP-P2A-Twin-Strep-GIGYF2 | This paper | | Human 4EHP and Twin-Strep tagged GIGYF2 for mammalian expression |
| Recombinant DNA reagent | pcDNA3.1-EDF1 (R85G)-TEV-3xFLAG | This paper | | Human EDF1 R85G mutant tagged with TEV-3x FLAG at the C-terminus for mammalian expression |
| Recombinant DNA reagent | pSP64-EDF1-TST | This paper | | Human EDF1 tagged with C-terminal TST for in vitro transcription and translation |
| Recombinant DNA reagent | pSP64-EDF1(R85G)-TST | This paper | | Human EDF1 R85G mutant with C-terminal TST for in vitro transcription and translation |
| Recombinant DNA reagent | pCMV-GFP-P2A-3XFLAG-VHPb-K(AAA)12+0-RFP | *Juszkiewicz and Hegde, 2017* | | Frameshifting reporters based on K12 poly sequence |
| Recombinant DNA reagent | pCMV-GFP-P2A-3XFLAG-VHPb-K(AAA)12+1-RFP | *Juszkiewicz and Hegde, 2017* | | Frameshifting reporters based on K12 poly sequence |
| Recombinant DNA reagent | pCMV-GFP-P2A-3XFLAG-VHPb-K(AAA)12–1-RFP | *Juszkiewicz and Hegde, 2017* | | Frameshifting reporters based on K12 poly sequence |
| Recombinant DNA reagent | pCMV-GFP-P2A-3XFLAG-VHPb-WT_XBP1_4AAA+0-RFP | This paper | | Frameshifting reporterts based on XBP1 stalling sequence |
| Recombinant DNA reagent | pCMV-GFP-P2A-3XFLAG-VHPb-S255A_XBP1_4AAA+0-RFP | This paper | | Frameshifting reporterts based on XBP1 stalling sequence |
| Recombinant DNA reagent | pCMV-GFP-P2A-3XFLAG-VHPb-W256A_XBP1_4AAA+0-RFP | This paper | | Frameshifting reporterts based on XBP1 stalling sequence |
| Recombinant DNA reagent | pCMV-GFP-P2A-3XFLAG-VHPb-WT_XBP1_4AAA+1-RFP | This paper | | Frameshifting reporterts based on XBP1 stalling sequence |
| Recombinant DNA reagent | pCMV-GFP-P2A-3XFLAG-VHPb-S255A_XBP1_4AAA+1-RFP | This paper | | Frameshifting reporterts based on XBP1 stalling sequence |
| Recombinant DNA reagent | pCMV-GFP-P2A-3XFLAG-VHPb-W256A_XBP1_4AAA+1-RFP | This paper | | Frameshifting reporterts based on XBP1 stalling sequence |

*Continued on next page*

Continued

| Reagent type (species) or resource | Designation | Source or reference | Identifiers | Additional information |
|---|---|---|---|---|
| Recombinant DNA reagent | pCMV-GFP-P2A-3XFLAG-VHPb-WT_XBP1_4AAA-1-RFP | This paper | | Frameshifting reporterts based on XBP1 stalling sequence |
| Recombinant DNA reagent | pCMV-GFP-P2A-3XFLAG-VHPb-S255A_XBP1_4AAA-1-RFP | This paper | | Frameshifting reporterts based on XBP1 stalling sequence |
| Recombinant DNA reagent | GFP-P2A-3XFLAG-VHPb-W256A_XBP1_4AAA-1-RFP | This paper | | Frameshifting reporterts based on XBP1 stalling sequence |
| Recombinant DNA reagent | pRSETA 6xHIS-TEV-eRF1(AAQ) | *Brown et al., 2015* | | Human mutant eRF1(AAQ) for expression in *E. coli* |
| Sequence-based reagent | CRISPR: EDF1 sgRNA1 | IDT | | 5' - ATCTTAGCGGCACAGAGACG - 3' |
| Sequence-based reagent | CRISPR: EDF1 sgRNA1 | IDT | | 5' - GAGCAAGGGGCTTACGCAGA - 3' |
| Sequence-based reagent | CRISPR: GIGYF2 sgRNA1 | IDT | | 5' - GGGAACACATGGAACGACGT - 3' |
| Sequence-based reagent | CRISPR: GIGYF2 sgRNA2 | IDT | | 5' - GGCGACTAGCTGGATCAAGG - 3' |
| Sequence-based reagent | siRNA: control | Thermo Fisher | 4390843 | Silencer Select |
| Sequence-based reagent | siRNA: EDF1 #1 | Thermo Fisher | #16610 | Silencer Select |
| Sequence-based reagent | siRNA: EDF1 #2 | Thermo Fisher | #s225027 | Silencer Select |
| Sequence-based reagent | siRNA: GIGYF2 #1 | Thermo Fisher | #s25033 | Silencer Select |
| Sequence-based reagent | siRNA: GIGYF2 #2 | Thermo Fisher | #s25034 | Silencer Select |
| Sequence-based reagent | siRNA: ZNF598 | Thermo Fisher | #s40509 | Silencer Select |
| Sequence-based reagent | siRNA: 4EHP | Thermo Fisher | #s18149 | Silencer Select |
| Sequence-based reagent | siRNA: 4EHP | Thermo Fisher | #s18150 | Silencer Select |
| Sequence-based reagent | GAPDH Forward primer for RT-PCR | IDT | | 5'-AGCTCATTTCCTGGTATGACA-3' |
| Sequence-based reagent | GAPDH Reverse primer for RT-PCR | IDT | | 5'-AGGGGAGATTCAGTGTGGTG-3' |
| Sequence-based reagent | RPLP1 Forward primer for RT-PCR | IDT | | 5'-CTCACTTCATCCGGCGACTAG-3' |
| Sequence-based reagent | RPLP1 Reverse primer for RT-PCR | IDT | | 5'-GCAGAATGAGGGCCGAGTAG-3' |
| Sequence-based reagent | GFP Forward primer for RT-PCR | IDT | | 5'-GGCAAGCTGACCCTGAAGTT-3' |
| Sequence-based reagent | GFP Reverse primer for RT-PCR | IDT | | 5'-CTTGTAGTTGCCGTCGTCCT-3' |
| Sequence-based reagent | RFP Forward primer for RT-PCR | IDT | | 5'-AGCAAGGGCGAGGAGGATAA-3' |
| Sequence-based reagent | RFP Reverse primer for RT-PCR | IDT | | 5'- TAGGCCTTGGAGCCGTACAT-3' |
| Peptide, recombinant protein | S7 Micrococcal Nuclease | Roche | Cat #11873580001 | purified protein |

*Continued on next page*

*Continued*

| Reagent type (species) or resource | Designation | Source or reference | Identifiers | Additional information |
|---|---|---|---|---|
| Chemical compound, drug | Hygromycin B | Millipore | Cat #400051-100KU | Selection antibiotic |
| Chemical compound, drug | Blasticidin S | Santa Cruz Biotechnology | Cat #sc204655 | Selection antibiotic |
| Chemical compound, drug | Emetine | Calbiochem | Cat #324693 | Used to induce ribosomal stalling |
| Software, algorithm | FlowJo | FlowJo | RRID:SCR_008520 | Analysis of FACS data |
| Software, algorithm | GraphPad Prism | GraphPad Prism | RRID:SCR_008520 | Statistical analysis, graphs |
| Other | Complete EDTA-free protease inhibitor cocktail | Roche | Cat #118735800001 | |

## Plasmids

The expression plasmid for 3xFLAG-tagged ZNF598 (pcDNA3.1-ZNF598-TEV-3xFLAG) was previously described (*Juszkiewicz and Hegde, 2017*) and is available at Addgene (#105690). The EDF1 expression plasmid (pcDNA3.1-EDF1-TEV-3xFLAG) was prepared the same way using the EDF1 ORF amplified from cDNA prepared from HEK293 cells. The R85G mutant of EDF1 was generated by site-directed mutagenesis. The EDF1-GFP plasmid was generated using Gibson assembly-based cloning with pcDNA3.1-EDF1-TEV-3xFLAG as a template and PCR amplified GFP-encoding sequence. The plasmid for co-expression of 4EHP and GIGYF2 was assembled in pcDNA3.1 as follows. First, a pcDNA3.1-based plasmid containing Twin-Strep-GIGYF2 was prepared, after which the 4EHP ORF [PCR amplified from a commercially available plasmid (Origene cat. RC201391)] was inserted 5' of a P2A sequence that preceded GIGYF2. Plasmids for in vitro transcription and translation of EDF1 were pSP64-based. EDF1 WT or R85G mutant ORF sequences were amplified from pcDNA-based plasmids and inserted 5' of the Twin-Strep tag (TST) resulting in pSP64-EDF1-TST and pSP64-EDF1(R85G)-TST plasmids. Previously described plasmid pRSETA-6xHIS-TEV-eRF1$^{AAQ}$ was used for recombinant expression of eRF1$^{AAQ}$ in *E. coli* (*Brown et al., 2015*). Plasmids for CRISPR-Cas9 gene disruption used px459 plasmid as a backbone (*Ran et al., 2013*), Addgene plasmid #62988), with sgRNA sequences that are provided in the Key resources table. Reporter constructs for frameshifting analysis at $(K^{AAA})_{12}$ sequences were previously described (*Juszkiewicz and Hegde, 2017*). Frameshifting reporters encoding WT, S255A and W256A XBP1 mutants were generated by standard restriction digestion and ligation of chemically synthesised oligonucleotides using $(K^{AAA})_{12}$ +0, +1,–1 frameshifting reporters as backbones.

## Cell lines

This work employed Flp-In T-REx 293 cells (from Thermo Fisher Scientific) and several stable cell lines derived from it. All cells are checked approximately monthly for mycoplasma contamination and verified to be negative. The identity of the founding cell line is verified by the presence of the FRT locus and accompanying antibiotic resistance. The derived cell lines are verified by antibiotic resistance and the reporters they express.

WT and ΔZNF598 Flp-In T-REx 293 cells with stably integrated, doxycycline-inducible reporters $(K^{AAA})_{21}$ and $(K)_0$ were previously described (*Juszkiewicz and Hegde, 2017*). ΔEDF1 and ΔGIGYF2 cells were generated using CRISPR-Cas9 methodology in Flp-In T-REx 293 cells already containing the $(K^{AAA})_{21}$ reporter. Briefly, the cells were transiently transfected with px459 plasmid encoding sgRNA sequences (see Key resources table). 24 hr post transfection, cells were transferred to media containing 1 µg/ml of puromycin to select the cells positively transfected with the plasmid. 72 hr post transfection, positively selected cells were trypsinised and subcloned into 96-well plates at a density close to 0.5 cell/well to obtain single cell clones. Colonies were allowed to grow for ~2 weeks. Individual clones were expanded and screened for the expression of protein using western blotting of the total cell lysates. Multiple clones (at least 3) showing no detectable expression of the

protein of interest were selected for further validation using phenotypic screening by flow cytometry.

## Cell cultures

All cells were maintained in Dulbecco's Modified Eagles's Medium (DMEM) with 10% of tetracycline-free fetal calf serum (FCS). In case of Flp-In T-REx 293 cells harbouring doxycycline inducible reporters, 15 µg/ml blasticidin and 100 µg/ml hygromycin were included in the media. Induction of the reporter expression was with 1 µg/ml of doxycycline for the times indicated in the figure legends. Treatment with emetine was for 1 hr with the following concentrations: high – 360 µM or low 1.8 µM unless otherwise indicated in the individual figure legends. Transfections with plasmid DNA were performed with TransIT-293 (Mirus) according to manufacturer's protocol. Silencing with siRNAs was typically for 72 hr (unless otherwise stated in the figure legends), using Lipofectamine RNAiMAX (Thermo Fisher). When longer silencing was needed, the cells were split and transfected again with fresh siRNA after the first 72 hr.

## Flow cytometry analysis

Analysis of the dual-fluorescent reporter using flow cytometry was as described previously (*Juszkiewicz and Hegde, 2017*). Briefly, cells cultured on 24-well plates, induced with doxycycline for the times indicated in the figure legends, were washed with PBS, trypsinised, collected in DMEM containing 10% FCS, and spun for 3 min at 5000 rpm using a tabletop centrifuge. Cell pellets were resuspended in 500 µl of ice-cold PBS, filtered through a 70 µm cell strainer, and analysed using the Becton Dickinson LSRII instrument. 20,000–30,000 GFP-positive events were collected for each sample. Data analysis used FlowJo software. For the analysis of frameshifting, Flp-In T-REx 293 cells cells were first transfected with siRNAs targeting EDF1 or non-targeting siRNAs as a negative control. After ~72 hr continued culture, cells were washed with PBS, trypsinised, and seeded again at lower density. Three parallel dishes were then each transfected with one of the three frameshifting constructs together with a BFP-encoding plasmid at a ratio of 1:1. Expression of the frameshifting reporters was allowed for ~20 hr before preparing the cells for flow cytometry analysis as described above. Data were collected on 20,000–30,000 BFP-positive cells (as a marker for transfection), with BFP-negative cells serving as a negative control to judge the level of background autofluorescence in each channel. To calculate the amount of frameshifting observed in +1 and −1 frames, median fluorescence intensity (hereafter MFI) of the RFP signal was derived using FlowJo software. The background RFP MFI derived from non-transfected cells was first subtracted from MFI values obtained for all the constructs, and the amount of signal in +1 and −1 frames was calculated as a percentage of the RFP MFI in +0 frame. GFP levels were verified to be equal among the three constructs as an indicator of equal expression.

## Immunoprecipitation experiments

The cell lines indicated in the figure legend were transfected with the constructs indicated in the figure legend. A single 10 cm dish was used for each experimental condition. Expression of transgenes was allowed for ~20 hr and where indicated in the figure, cells were additionally treated with 1.8 µM emetine for 30 min immediately prior to cell lysis. Each plate was washed with ice cold PBS twice, cells were collected in ice cold PBS by scraping, and spun for 3 min at 5,000 rpm using a tabletop centrifuge chilled to 4°C. For the experiments in *Figure 5A, B and D*, lysis was performed for 15 min on ice using 400 µl of 1xRNC buffer supplemented with 0.5% Triton X-100 and 1 mM DTT. The nuclei and cell debris were spun for 10 min at 15,000 g using a tabletop centrifuge at 4°C. The postnuclear supernatant was further spun for 1 hr at 100,000 rpm at 4°C using a TLA120.2 rotor to obtain ribosome-free cytosolic extracts. FLAG-M2 resin or GFP-trap agarose resin were added directly to the ribosome-free supernatants and incubated for 1 hr at 4°C with end-over-end rolling. Beads were washed with the lysis buffer four times and elution was performed for 5 min at 95°C using SDS-sample buffer (in case of the GFP trap resin) or for 45 min at room temperature with shaking using 0.2 mg/ml FLAG peptide in 1xRNC. For the experiment in *Figure 5C*, cells were lysed for 15 min on ice using 200 µl of 1xRNC supplemented with 0.05% digitonin and 1 mM DTT. Cell rupture was ensured by passing the lysates through 26G needle appended to 1 ml syringe. The lysates were spun for 10 min at 15,000 g and supernatants were loaded on 2 ml of pre-equilibrated 10–50% sucrose

gradients. The gradients were centrifuged at 55,000 rpm for 1 hr at 4°C using the TLS-55 rotor and slowest acceleration and deceleration settings. 11 fractions, 200 µl each were collected manually from the top of the gradient and fractions 1–4 (non-ribosomal fractions), and fractions 7–10 (heavy polysomal fractions) were pooled together. Digitonin was added to each of the pooled fractions to achieve 0.05% final concentration and the IP was performed for 1 hr in the cold room with end-over-end rolling using anti-FLAG M2 affinity resin. The resin was washed four times with the lysis buffer and elution was performed as described above using 0.2 mg/ml FLAG peptide in 1xRNC buffer.

## Quantitative mass spectrometry of purified polysomes

HEK293 cells were grown on 10 cm plates. One 10 cm plate was used for each experimental condition, which was performed in duplicate. 90% confluent cells were treated for 15 min with a low dose of emetine (1.8 µM), a high dose of emetine (360 µM) or with DMSO (control). Prior to harvesting, cells were washed with ice cold PBS twice. Lysis was in 200 µl for 20 min on ice using 1xRNC buffer [50 mM HEPES pH 7.6, 100 mM K(OAc), 5 mM Mg(OAc)$_2$] supplemented with 0.01% purified digitonin, 40 U/ml RNAsin, 1x protease inhibitor cocktail, and 1 mM DTT. Cells were ruptured by passing through a 26G needle using a 1 ml syringe (20 passes). Lysates were sedimented by centrifugation for 15 min at 15,000 g at 4°C using a tabletop centrifuge. The resulting supernatant was loaded onto a 10–50% sucrose gradient (2 ml gradient volume) in 1xRNC buffer. Centrifugation was for 1 hr at 55,000 rpm in the TLS 55 rotor (Beckman) at 4°C using the slowest acceleration and deceleration settings. After the spin, 11 fractions were manually collected from the top of the gradient. Fractions 8–11 were pooled together (previously verified to represent polysomal fractions), diluted with an equal volume of 1xRNC buffer (800 µl) and centrifuged for 1 hr at 100,000 rpm in the TLA100.3 rotor at 4°C. The polysome pellet was washed once with 200 µl of 1xRNC and resuspended in 20–50 µl of 1xRNC buffer. The concentrations of samples were normalised using absorbance at 260 nm and subjected to quantitative mass spectrometry using Tandem Mass Tagging (TMT) approach (*Thompson et al., 2003*). In brief, quantitative mass spectrometry was performed by in-solution digestion of samples, followed by tandem mass tag labelling (Thermo Fisher Scientific cat. #90110). Protein samples in solution were reduced with 5 mM DTT at 56°C for 30 min and alkylated with 10 mM iodoacetamide for 30 min in the dark at 22°C. The alkylation reaction was quenched by the addition of DTT and the samples were digested with trypsin (Promega, 0.5 µg) overnight at 37°C. The peptide mixtures were then desalted using home-made C18 (3M Empore) stage tip filled with 0.5 mg of poros R3 (Applied Biosystems) resin. Bound peptides were eluted sequentially with 30%, 50% and 80% acetonitrile in 0.1%TFA and lyophilised. Dried peptide mixtures from each condition were resuspended in 20 µL of 7% acetonitrile, 200 mM triethyl ammonium bicarbonate. For TMT labelling, 0.8 mg of TMT reagents (Thermo Fisher Scientific) were reconstituted in 41 µL anhydrous acetonitrile. 10 µL of TMT was added to each peptide mixture and incubated for 1 hr at room temperature. The labelling reactions were terminated by incubation with 2.5 µL 5% hydroxylamine for 15 min, and labelled samples were subsequently pooled. The acetonitrile was evaporated using a Speed Vac, desalted and then fractionated with home-made C18 (3M Empore) stage tip using 10 mM ammonium bicarbonate and acetonitrile gradients. Eluted fractions were partially dried down using a Speed Vac and subjected to LC-MSMS. Liquid chromatography was performed on a fully automated Ultimate 3000 RSLC nano System (Thermo Scientific) fitted with a 100 µm × 2 cm PepMap100 C18 nano trap column and a 75 µm × 25 cm reverse phase C18 nano column (Aclaim PepMap, Thermo Scientific). Samples were separated using a binary gradient consisting of buffer A (2% acetonitrile, 0.1% formic acid) and buffer B (80% acetonitrile, 0.1% formic acid). Peptides were eluted at 300 nL/min with an acetonitrile gradient. The HPLC system was coupled to a Q Exactive Plus hybrid quardrupole-Orbitrap mass spectrometer (Thermo Fisher Scientific) equipped with a nanospray ion source. The acquired MSMS raw files were processed using Proteome Discoverer (version 2.1, Thermo Scientific). MSMS spectra were searched against mammals, UniProt Fasta database using Mascot (version 2.4, Matrix Science) search engine.

## Analytical sucrose gradient fractionations

Small scale, analytical sucrose gradient fractionations were as described previously (*Juszkiewicz and Hegde, 2017*). Briefly, 20 µl of sample was loaded on 200 µl of 10–50% sucrose gradient prepared by overlaying 40 µl layers of 50%, 40%, 30%, 20% and 10% sucrose solutions dissolved in 1xRNC

buffer. Gradients were allowed to equilibrate for 1 hr prior to centrifugation. Spin was for 20 min, at 4°C, at 55,000 rpm using slowest acceleration and deceleration settings in TLS-55 rotor (Beckman). Eleven consecutive fractions, 20 µl each, were collected manually from the top of the gradient and analysed directly.

### Analysis of cell lysates using sucrose gradient fractionation

Cells at ~80% confluency, grown on 10 cm plates (two plates per sample) were either treated with low dose (1.8 µM) of emetine or DMSO control for 1 hr. Before harvesting by scraping, cells were moved on ice and washed twice with ice cold PBS. Harvested cells were spun at 4°C at 1000 rpm for 5 min and resuspended in 100–200 µl of 1xRNC buffer containing 0.01% purified digitonin, 40 U/ml RNAsin inhibitor (Promega), 1 mM DTT, and 1x EDTA-free protease inhibitors cocktail. After 15 min incubation on ice, cells were ruptured by passing through pre-chilled 21G needle attached to 1 ml syringe ~15 times. Cell debris was removed by 10 min spin at 15,000 g at 4°C using a tabletop micro-centrifuge. The protein concentrations of post-spin supernatants were measured using nanodrop. 150 µg of protein in the lysate was adjusted to 20 µl volume using lysis buffer, loaded on 10–50% analytical sucrose gradient and analysed as described above.

### Western blotting

Western blotting was performed as described previously (*Juszkiewicz and Hegde, 2017*). For the analysis of cell-derived samples, cells were washed twice with ice cold PBS and harvested. After short spin (3 min at 5000 rpm) in microcentrifuge, cell pellets were lysed using 100 mM Tris pH 8.0 with 1% SDS. To shear genomic DNA, lysates were boiled at 95°C and vortexed extensively. Samples were analysed using Tris-Tricine SDS-PAGE electrophoresis followed by wet transfer to 0.2 µm nitro-cellulose membrane. Blocking was for 1 hr at room temperature and incubation with primary anti-bodies was for 1 hr at room temperature or for 16 hr at 4°C. For detection, we used HRP-conjugated secondary antibodies and SuperSignal West Pico Chemiluminescent substrate.

### In vitro transcription and translation

In vitro transcription and translation reactions were performed as described previously in detail (*Sharma et al., 2010*). In vitro transcription used 5–20 ng/µl of purified PCR product in 40 mM HEPES pH 7.4, 6 mM MgCl$_2$, 20 mM spermidine, 10 mM reduced glutathione, 0.5 mM ATP, 0.5 mM UTP, 0.5 mM CTP, 0.1 mM GTP, 0.5 mM CAP, 0.4–0.8 U/µl RNAsin and 0.4 U/µl SP6 polymerase and was performed for 1 hr at 37°C water bath. In vitro translation was performed using two rabbit reticulocyte lysate (RRL) systems: non-nucleased RRL was used to generate ribosomes collided at the stop codons of endogenous mRNAs (mostly α- and β-globin) and RRL treated with micrococcal nuclease was used to translate WT and mutant EDF1 for purification. Translation reactions contained 33% by volume crude RRL (Green Hectares), 0.5 µCi/µl $^{35}$S-methionine, 20 mM HEPES, 10 mM KOH, 2 mM MgCl$_2$, 40 µg/ml creatine kinase, 20 µg/ml pig liver tRNA, 12 mM creatine phosphate, 1 mM ATP, 1 mM GTP, 50 mM K(OAc), 2 mM MgCl$_2$, 1 mM reduced glutathione, 0.3 mM spermidine and 40 µM of each amino acid except methionine. Each translation also contained mock (in case of non-nucleased RRL) or template-containing transcription reaction (nucleased RRL) which constituted 5% of the reaction by volume. Translations were for 20–45 min in a 32°C water bath.

### Recombinant eRF1$^{AAQ}$ production in *E. coli*

Recombinant eRF1$^{AAQ}$ was produced as described before (*Brown et al., 2015*) with minor modifica-tions. In brief, *E. coli* strain BL21 DE3 pLysS was transformed with pRSETA/His$_6$-TEV-eRF1$^{AAQ}$ and grown on ampicillin-containing plates. Two flasks each containing 1 L ampicillin-containing LB were inoculated with 30% of the bacterial lawn from the plate and grown until 0.6 OD was reached. Induc-tion of recombinant protein production was with 0.2 mM of IPTG for 2 hr at 37°C. Cells were sedi-mented by 30 min centrifugation in JA8.1 at 6000 rpm, washed once with 25 ml of ice-cold PBS buffer and spun again as above. Cell pellets were flash frozen in liquid nitrogen. For purification, cell pellets were thawed in room temperature water bath. Lysis was performed in 1xPBS supplemented with 250 mM NaCl, 10 mM imidazole, 1 mM DTT, 1x protease inhibitors cocktail by sonication. Lysate was precleared by 40 min spin at 18 000 rpm in JA 25.50 rotor and supernatant was applied to pre-equilibrated 1 ml of NiNTA resin (Qiagen). Resin was washed with 50 ml of lysis buffer and

eluted with three bed volumes of elution buffer: 1xPBS, 250 mM NAcl, 250 mM imidazole, 1 mM DTT. The two elution fractions (2 ml) were pooled and dialysed against 1 L of dialysis buffer overnight (1xRNC, 10 mM imidazole, 10% glycerol, 1 mM DTT) in the presence of TEV protease in 1/50 ratio (weight/weight protein). Dialysate was spun for 15 min at max speed in tabletop microcentrifuge to remove any precipitate. Supernatant was applied to 1 ml of pre-equilibrated NiNTA resin to deplete any protein with uncleaved tag. Aliquots of the untagged eRF1$^{AAQ}$ were flash frozen and stored in −80°C.

## RT-qPCR

Flp-In T-REx 293 cells containing the (K$^{AAA}$)$_{21}$ stalling reporter at the FRT site were transfected with siRNAs as described in the figure legend. After ~72 hr, cells were induced for 6 hr with 1 µg/ml of doxycycline and harvested on ice. Total RNA extraction was performed using RNeasy Mini kit (Qiagen) according to manufacturer's protocol. Additionally, on column DNA digestion was perform to digest genomic DNA. 500 ng of total RNA was used to generate cDNA libraries using iScript cDNA synthesis kit (Biorad). Each sample was diluted 10-fold with nuclease free-water. Small quantities of all samples were pooled together and subsequently serially diluted to generate a standard curve. Each sample was additionally diluted 10-fold to ensure concentration within a standard curve. qPCR was performed using Viia 7 Real-Time PCR System (Thermo Fisher) and KAPA SYBR Fast qPCR reagents (KAPA Biosystems) according to manufacturer's protocol. The primer sequences are listed in the Key resources table. All pairs were annealed at 60°C and a melt curve was performed. Data was analysed using Quantstudio Real-Time PCR software v1.3. Values are normalised to the standard curve and against RPLP1 or GAPDH reference genes. Experiments include two biological replicates, each containing three technical replicates performed on two different sets of primers targeting the same transcript. Non parametric pairwise Mann-Whitney test was performed using Graphpad (Prism) software to asses statistical significance of the observed differences in the relative mRNA levels.

## Acknowledgements

We thank M Skehel and M Daly for maintaining the mass spectrometry and flow cytometry facilities, respectively; S Kraatz, V Chandrasekaran, and M Babu for useful discussions; M Höpfler for comments on this manuscript. This work was supported by the UK Medical Research Council (MC_UP_A022_1007 to RSH).

## Additional information

### Competing interests

Ramanujan S Hegde: Reviewing editor, *eLife*. The other authors declare that no competing interests exist.

### Funding

| Funder | Grant reference number | Author |
|---|---|---|
| Medical Research Council | MC_UP_A022_1007 | Ramanujan S Hegde |

The funders had no role in study design, data collection and interpretation, or the decision to submit the work for publication.

### Author contributions

Szymon Juszkiewicz, Conceptualization, Investigation, Writing - original draft, Performed all experiments displayed in this study and co-wrote the manuscript; Greg Slodkowicz, Data curation, Formal analysis, Writing - review and editing, Provided analysis of a ribosome profiling experiment that influenced interpretation of this study, but was not included in the final revised manuscript; Zhewang Lin, Conceptualization, Investigation, Writing - review and editing, Prepared samples for a ribosome profiling experiment that influenced interpretation of this study, but was not included in the final revised manuscript; Paula Freire-Pritchett, Data curation, Formal analysis, Writing - review and editing,

Helped to analyze a ribosome profiling experiment that influenced interpretation of this study, but was not included in the final revised manuscript; Sew-Yeu Peak-Chew, Investigation, Writing - review and editing, Performed and help analyze mass spectrometry experiments; Ramanujan S Hegde, Conceptualization, Supervision, Funding acquisition, Writing - original draft, Project administration

### Author ORCIDs
Szymon Juszkiewicz (iD) http://orcid.org/0000-0002-3361-7264
Greg Slodkowicz (iD) https://orcid.org/0000-0001-6918-0386
Ramanujan S Hegde (iD) https://orcid.org/0000-0001-8338-852X

### Decision letter and Author response
Decision letter https://doi.org/10.7554/eLife.60038.sa1
Author response https://doi.org/10.7554/eLife.60038.sa2

## Additional files

### Supplementary files
• Transparent reporting form

### Data availability
All data generated or analyzed during this study are included in the manuscript and supporting files. Source data have been provided for Figure 1. The mass spectrometry proteomics data have been deposited to the ProteomeXchange Consortium via the PRIDE partner repository with the dataset identifier PXD020239.

The following dataset was generated:

| Author(s) | Year | Dataset title | Dataset URL | Database and Identifier |
|---|---|---|---|---|
| Juszkiewicz S, Slodkowicz G, Lin Z, Freire-Pritchett P, Peak-Chew SY, Hegde RS | 2020 | Ribosome collisions trigger cis-acting feedback inhibition of translation initiation | https://www.ebi.ac.uk/pride/archive/projects/PXD020239 | PRIDE, PXD020239 |

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
