## [Decision Letter]

**Acceptance summary:**

This article suggests a hitherto elusive mechanism implicated in resolving ribosome collisions that appears to be distinct from the previously described quality control mechanism wherein aborted translation is sensed by ZNF598. To this end, the authors provide evidence that recruitment of EDF1 to collided ribosomes triggers inhibition of translation initiation in cooperation with GIGYF2 and 4EHP. Future studies are warranted to establish the potential physiological role(s) of this EDF1-driven response to ribosome stalling.

**Decision letter after peer review:**

[Editors’ note: the authors submitted for reconsideration following the decision after peer review. What follows is the decision letter after the first round of review.]

Thank you for submitting your work entitled "Ribosome collisions trigger cis-acting feedback inhibition of translation initiation" for consideration by *eLife*. Your article has been reviewed by three peer reviewers, including Ivan Topisirovic as the Reviewing Editor and Reviewer #1, and the evaluation has been overseen by a Senior Editor.

Our decision has been reached after consultation between the reviewers. Based on these discussions and the individual reviews below, we regret to inform you that your work will not be considered further for publication in *eLife* at this stage.

There was general enthusiasm pertinent to the potential role of the EDF1-GIGYF2-4EHP axis in sensing and resolving ribosome collisions. Unfortunately, however, the reviewers felt that the ribosome profiling studies suffered from insufficient statistical rigor due to the lack of biological replicates. In addition, it was found that the mechanistic data corroborating the latter model were somewhat preliminary and/or open to alternative explanations. Specifically, it was thought that the analysis of ribosome profiling data should be improved by including independent biological replicates and statistically rigorous methodology. This should be complemented by orthogonal validation of ribosome profiling data and ZNF598-dependent stalls. In addition, experiments corroborating the proposed model would also benefit from sufficient replication. Important controls outlined in each of the reviewer's comments should also be included. Addressing issues raised regarding eS10 ubiqutination as well as the potential competition between GIGYF2-4EHP binding to EDF1 vs. ZNF598 would also be appreciated. The latter experiments should be carried out using more direct approaches (e.g. co-IP). Finally, excluding and/or considering alternative explanations pertinent to studies employing emetine was also thought to be warranted.

We realize that this news will be disappointing. If you plan to address these issues and resubmit this study to *eLife* as a new submission, we will try to expedite the review process by sending the manuscript to the same reviewers that reviewed the initial submission and include your responses to the reviewer comments from this round of evaluation. We hope you will find these comments constructive in planning your next steps.

Reviewer #1

In this study, Juszkiewicz et al. provide evidence suggesting a previously unappreciated mechanism of resolving ribosome collisions centred on EDF1. This mechanism appears to be distinct form the quality control mechanism wherein aborted translation is sensed by ZNF598. To this end, the authors provide evidence that recruitment of EDF1 to collided ribosomes may inhibit translation initiation via recruitment of GIGYF2 and 4EHP. Based on these data, it is proposed that EDF1 examines mRNAs for increased ribosome density, followed by the recruitment of GIGYF2 to block initiation and facilitate resolving of ribosome collisions. Failure of this mechanism is suggested to be followed by induction of ZNF598-directed mechanism. Overall, it was found that this study is of broad potential interest as it describes a hitherto underexplored mechanism of resolution of ribosomal stalling and collisions. Notwithstanding general enthusiasm for the model, it was found that this study falls short when it comes to including a number of important controls as well as biological replicates to support authors' conclusions. Specific comments are outlined below:

Major comments:

– Several major issues were observed concerning inadequate scientific rigor, including insufficient number of replicates. It is stated in the legend of Figure 1—figure supplement 1 that ribosome profiling experiments were done in 2 technical replicates, which, if not a typo, suggest a single biological replicate. This precludes any meaningful statistical analysis. It is also expected that high reproducibility will be obtained from technical replicates, and it is thus not clear why technical replicates were used en lieu of biological replicates.

– A number of additional issues were noted pertinent to ribosome profiling analysis:

"To determine whether a subset of mRNAs are more dependent on ZNF598 than others, we tabulated the KO/WT ratio of CFr for each ORF. This value was plotted relative to that ORF's total dRPFs as an indicator of sampling depth, and hence reliability of the measurement (Figure 1C). 339 ORFs were higher (defined as ZNF598-dependent ORFs) and 159 ORFs lower than a significance interval defined by two standard deviations flanking a rolling mean".

This is lacking any statistical rigor as there is no adjustment for multiple testing. It would be more appropriate to perform biological replicates and to statistics on these.

"When normalized to the number of background rRNA reads as an internal control..."

Using rRNA to normalize is highly inadvisable as rRNA typically varies for a number of reasons. It also seems that the way rRNA reads are used to normalize the data, does not affect dRPFs to mRPFs ratio.

Figure 1—figure supplement 2: My opinion is that the correct approach here is to establish whether there are more dRPFs when TE adjusted for mRNA length. To this end, Y axis should represent dRPFs, while X axis should represent translational efficiencies calculated as residuals from a regression between mRPFs and mRNA levels (reads only from the coding region should be used). This will adjust for both length of the mRNA and the expression level.

"One explanation for the absence of a correlation is if collisions on highly translated mRNAs are efficiently resolved by ZNF598-triggered disassembly. However, a matched ribosome profiling dataset from ZNF598 knockout (KO) cells showed a similar range of CFr and no correlation with TE (Figure 1—figure supplement 2A, orange dots). This indicates that highly translated mRNAs (as defined by high TE) neither show a higher than average basal collision frequency nor are they preferentially affected in ZNF598 KO cells"

This seems as over-interpretation given how these effects were calculated, and the inherent noisiness of the assay. To make this claims RUST analysis should also be performed.

"…would be more likely to occur on mRNAs with high ribosome density."

It seems equally plausible that local features of the CDS may explain this phenomenon.

"Due to normalization, the above analysis cannot inform about any uniform and wholesale differences in collision frequency between WT versus KO cell"

The authors should corroborate this statement as their data indicate largely uniform effects.

"As exemplified by HNRNPU and RPS4X, some ORFs that did not score as hits when averaging CFr across the ORF nevertheless showed individual collision site(s) that are strongly dependent on ZNF598 (Figure 2C)."

How were these identified? Do they occur more than by chance?

"Such examples indicate that disome profiling is capable of identifying highly specific and strongly ZNF598-dependent stall sites with codon precision".

Seems overstated without any systematic analysis and validation that these indeed represent true events.

"Third, the collision frequency is remarkably similar across a wide range of translation efficiencies"

Data to support this statement were found to be insufficient.

Several other weaknesses were observed:

– "In addition to ZNF598 as the most collision-specific interaction partner, we identified eight other candidate collision-specific proteins."

Can authors explain how were these proteins identified as candidates?

– "Only two of the candidates, EDF1 and GIGYF2, were collision-specific interactors in both WT and ZNF598 KO cells"

How was this calculated/determined?

– Figure 3D – Western blot indicating the extent of ZNF598 depletion is missing.

Reviewer #2

Ribosome stalling and collisions, as well ribosome-associated quality control have drawn considerable attention over the last several years. It was postulated that major substrate for this ribosome rescue pathway are ribosomes stalled on damaged and prematurely polyadenylated mRNAs. The stalls are resolved by no-go decay factors, Pelota-Hbs1, and ZNF598(Hel 2), however there are still gaps in the mechanism of this process. In this manuscript, Juszkiewicz and colleagues, demonstrate an interesting observation that the ribosome collisions may initiate translation initiation inhibition in cis independently of ZNF598 pathway.

Unfortunately, despite the clear support that EDF1 is involved in ribosome collisions and resolution of this process, most of the claims are not clearly or fully supported by the presented data.

1) The complete analyses of ribosome profiling data and dRPF/mRPF ratio has multiple limitations. It is not clear whether these data are analyzed for the sequencing bias and normalization by dividing both sets with same rRNA background reads is problematic. The three conclusions drawn from these data are as such not supported.

2) Authors never report that experiments shown in Figure 3C and Figure 5B are from the same set of data, but they clearly use the same blots in these two figures. Have authors did replicates of these experiments?

3) It is not completely clear how eS10 gets ubiquitinated by ZNF598 in cells missing GIGYF2 (Figure 6). The supplement Figure 6 shows no recruitment of ZNF598 to poly-ribosomes in those same conditions with a rather massive recruitment in case of the WT cells. Authors argue that this could be due to transient interaction of ZNF598 in cells missing either EDF1 or GIGYF2. Is es10 ubiquitinated in HEK293 cells missing ZNF598, or Caco-2 and MCF-7 cells?

4) Original report by Morita et al., 2012. has 4EHP-GIGYF2-ZNF598 complex as translational repressor with direct co-IPs between all components and especially between 4EHP and ZNF598 as well as GIGYF2 and ZNF598. Is EDF1 binding to GIGYF2-4EHP and competing with ZNF598? There is no co-IP studies shown on any components so the reader is left to believe on interactions through the polysome profiles or functional assays.

Reviewer #3

The authors present a study on the dynamics of ribosome collisions and how modulations of the RQC pathway aim to resolve such aberrant events. By using a combination of disome-seq to identify collisions, clever reporter constructs and unbiased proteomics they shed light on different branching of the RQC pathway. Despite some important shortcomings pointed out in the following comments, the methods and conclusions are sound, the paper is well written, and it represent a strong contribution to the field.

1) Steady-state RNA abundance in the different tested conditions has not been measured, making it impossible to appreciate possible changes in translation efficiency for the transcripts displaying collision events. The authors should show that such concern is not valid (the plot in Figure 2B does not represent a sufficient assessment), by experimental methods or analysis of other datasets. If not possible, the Authors should acknowledge this significant shortcoming, and revisit some of their conclusions where necessary.

2) The authors should compare their findings with recent work outlining the crosstalk between translation initiation and the RQC: https://www.biorxiv.org/content/10.1101/792994v1

3) The authors might want to explore the potential sequence/structure features underlying pausing collisions events and their potential in discriminating between ZNF598-dependent and other pausing events. Results from recent work on detecting pausing events should be considered (e.g. https://www.biorxiv.org/content/10.1101/746875v2 , https://www.biorxiv.org/content/10.1101/710061v1 , https://www.biorxiv.org/content/10.1101/710491v1).

4) In the emetine treated samples, the percentage of dRPF reads mapping to the ORF greatly increases. Do the authors observe robust pausing/collisions outside coding regions in the different conditions, or do they represent possible contaminants?

5) Figure 3C shows how both EDF1 and GIGYF2 migrate to the polysome fraction under emetine treatment. However, this might represent a more general phenomenon where many more protein complexes migrate to the polysome fraction. Can the author comment on this possibility?

[Editors’ note: further revisions were suggested prior to acceptance, as described below.]

Thank you for resubmitting your work entitled "Ribosome collisions trigger cis-acting feedback inhibition of translation initiation" for further consideration by *eLife*. Your revised article has been evaluated by Suzanne Pfeffer (Senior Editor) and a Reviewing Editor.

The manuscript has been improved but there are some remaining textual issues that need to be addressed before acceptance, as outlined below:

1) It was thought that the authors should either tone down some statements or discuss alternative explanations of observed phenomena. In particular, the interpretation that EDF1 recruits GIGYF2 complex at collided ribosomes was found to be insufficiently supported by data. To this end, it was found that although the IP experiments support the role of EDF1 in stabilization of GIGYF2 complex binding to collided ribosomes, their role in its recruitment remains unclear. This should be clearly stated throughout the manuscript and depicted in the model shown in Figure 5. It was also thought that Figure 7 should be removed as this model would require further kinetic studies that, as we agree, are out of the scope of this study.

2) Although highly appreciated, several issues were raised regarding frame-shifting studies. For instance, it appears that EDF1 depletion favours -1 frameshifting which cannot be fully explained by the collision model. Moreover, the authors cite studies that indicate that frameshifting should be in +1 direction. Discussion of these apparent discrepancies appears to be warranted. Potential effects of frameshifting on NMD activation should also be discussed. Finally, RFP signal and RFP/GFP ratios should be presented along with GFP for EDF1 knockdown and KO cells (Figure 4 and Figure 4—figure supplement 1) to further facilitate interpretation of results in Figure 6 and Figure 6—figure supplement 3.

3) It was also thought that the authors should specify number of replicates and precise criteria for some statistical tests. For example, 1.6 fold threshold was found to be quite arbitrary, and the authors should clearly elaborate on criteria to use such cutoff. MS data should be made publicly available and identified factors that are enriched/depleted on collided ribosomes should be at least briefly discussed. The authors are are also encouraged to add all appropriate controls (e.g. showing that ZNF598 is indeed knocked out in the experiment in question rather than referring to previous publications where these lines were established)

4) At times, interpretation of data was thought to require some clarification. Particularly, Figure 2—figure supplement 1A shows apparent increase in mRNA levels, albeit in the text the authors state that there is a lack of the potential mRNA abundance changes. To this end, it was thought that appropriate significance testing is required to corroborate the authors' conclusions, and if it turns out that the changes in mRNA abundance are significant, this should be commented on. Moreover, it is argued that 4EHP, but not GIGYF2 depletion increases translation. However, considering the scale in Figures 2C vs. 2D, it appears that the changes are of comparable amplitude but in the opposite directions. Based on this, significance testing appears warranted.

5) The authors are advised to verify some of the references. For instance in Mills et al., 2016 study does not appear to demonstrate ZNF598 loss during maturation of reticulocytes.

---

## [Author Response]

[Editors’ note: the authors resubmitted a revised version of the paper for consideration. What follows is the authors’ response to the first round of review.]

We appreciate the overall enthusiasm for our discovery of an EDF1-GIGYF2-4EHP mechanism of collision-specific translation repression. A large majority of the concerns related to inadequate analysis of a ribosome profiling experiment that aimed to provide a survey of collisions across the transcriptome. Because this experiment is tangential to the key findings of a new collision dependent response, we and the handling editor agreed that the study would be better if the profiling was omitted and the rest of the manuscript strengthened. In the response below, we address the non-profiling comments; the comments relating to profiling are no longer relevant.

Key issues identified by the Editor:1) Is EDF1 binding to GIGYF2-4EHP and competing with ZNF598? Moreover, what is the status of GIGYF2 complexes (with ZNF598 vs. EDF1) in emetine treated vs. control cells? Is ZNF598GIGYF2-4EHP still preserved in EDF1 knockout cells? Is interaction between ZNF598 and 4EHP (Morita et al., 2012, MCB) affected in GIGYF2 knockout cells? IP experiments were thought to be an appropriate approach here.

Using co-IP experiments, we examined the potential interactions between EDF1, GIGYF2, and ZNF598 to complement previous experiments by Morita et al., 2012. The IPs were performed from the ribosome-free cytosolic fraction to avoid co-precipitation due to bridging via polysomes. As shown in the new Figure 5, we find that ZNF598 interacts with GIGYF2, but that EDF1 does not interact with either. The absence of EDF1 interaction was verified with two different tags. The ZNF598-GIGYF2 interaction is retained in EDF1 knockout cells, further arguing that EDF1 is not an integral part of this complex away from the ribosome. Earlier work established that the ZNF598 interaction with 4EHP is via GIGYF2 (Figure 5B in Morita et al., 2012). Here, mutation of the 4EHP interaction motif in GIGYF2 prevented co-IP of 4EHP by ZNF598 despite recovery of GIGYF2. Because the GIGYF2-4EHP interaction is well established and validated with mutants, it was not re-explored in our experiments. Collectively, these findings show that free in the cytosol, ZNF598 interacts with GIGYF2, which interacts with 4EHP. This complex does not appear to include or require EDF1.

Interactions among these proteins on the ribosome cannot be deduced by co-IP because the ribosome and/or mRNA may bridge proteins that are nowhere close to each other. Nevertheless, we can observe by co-fractionation that EDF1, GIGYF2 (Figure 1B), and ZNF598 (Figure 6—figure supplement 2) are all selectively on collided ribosomes. While this finding by itself does not demonstrate they are all together on the same collided ribosome, this is a likely interpretation for the following reasons:

i) GIGYF2 interacts with ZNF598 and 4EHP off the ribosome (Figure 5, as described above), indicating that they are probably recruited together to collisions.

ii) Recruitment of GIGYF2 to collisions is impaired without EDF1 (Figure 4B) arguing that GIGYF2 is recruited to ribosomes that contain EDF1.

iii) Recruitment of ZNF598 to collisions is impaired without GIGYF2 or EDF1 (Figure 6—figure supplement 2) arguing that ZNF598 is recruited to ribosomes containing these factors.

Based on the collection of these data, together with the finding that EDF1 recruitment to collisions does not depend on ZNF598 (Figure 1C) or GIGYF2 (Figure 5—figure supplement 1), we conclude that EDF1 engages collided ribosomes and facilitates the recruitment of pre-formed ZNF598GIGYF2-4EHP complexes (see diagram in Figure 5E). The spatial arrangement of this complex on the ribosome will require structural analysis that is beyond the scope of this study.

2) How is eS10 ubiquitinated by ZNF598 in cells missing GIGYF2 (Figure 6)? The supplement Figure 6 shows that ZNF598 is not recruited to polysomes in the latter cells. It is argued that this may be due to transient interaction of ZNF598. However, it was thought that direct experimental evidence corroborating this tenet should be provided.

Long exposures of the immunoblots show that although weak, ZNF598 does interact selectively with collided ribosomes even in cells lacking GIGYF2 or EDF1 (Figure 6—figure supplement 2). This is consistent with a transient interaction mediating the observed eS10 ubiquitination. To verify this conclusion and exclude the possibility that another ligase ubiquitinates eS10, we analysed eS10 ubiquitination upon ZNF598 depletion in wild type cells and those lacking GIGYF2 (Figure 6C). The complete loss of ubiquitinated eS10 in each case demonstrates that ZNF598 is the sole E3 ligase for eS10, arguing that indeed a weak interaction is sufficient for ubiquitination to proceed effectively. This is now explained in the revised text that accompanies the above new experiments.

3) MbF1 (yeast EDF1 homolog) is thought to prevent frameshifting in yeast cells on stalled ribosomes (Wang et al., 2018). Is there more frameshifting with reporters in EDF1-depleted Hek293 cells relative to controls?

We analyzed this in some detail using four different reporters: (i) a weakly stalling poly(A) sequence; (ii) the Xbp1 stalling sequence; (iii) a strong-stalling variant of Xbp1; (iv) a weakstalling variant of Xbp1. As detailed in Figure 3 and its accompanying two supplements, depletion of EDF1 results in increased frameshifting at the Xbp1 stall and its strong-stalling variant, but not at the weaker stalling Xbp1 mutant or the weakly stalling poly(A) sequence. As discussed in the relevant parts of the text, we conclude that stalling and the ensuing ribosome collisions cause frameshifting (as in yeast), and that EDF1 mitigates this similar to Mbf1 in yeast (although there are clearly differences, as discussed in the text).

Reviewer #1– "In addition to ZNF598 as the most collision-specific interaction partner, we identified eight other candidate collision-specific proteins."Can authors explain how were these proteins identified as candidates?– "Only two of the candidates, EDF1 and GIGYF2, were collision-specific interactors in both WT and ZNF598 KO cells"How was this calculated/determined?

Sorry for not explaining this. We have now amended the text to explicitly indicate how many total proteins were identified in the mass spectrometry experiment (~600), what criteria were used to identify candidates (1.6-fold enrichment on collided polysomes relative to both untreated and fully stalled polysomes), and what further criteria led us to EDF1 and GIGYF2 (1.6-fold enrichment on collided ribosomes even in ZNF598 knockout cells).

– Figure 3D – Western blot indicating the extent of ZNF598 depletion is missing.

We apologise for the confusion. These are CRISPR-mediated knockout cells that were previously validated to have no detectable ZNF598 (Juszkiewicz and Hegde, 2017).

Reviewer #2Authors never report that experiments shown in Figure 3C and Figure 5B are from the same set of data, but they clearly use the same blots in these two figures. Have authors did replicates of these experiments?

We apologise for this oversight. In a previous draft of this manuscript, Figure 3C and 5B were part of a single figure from an experiment that was later presented in two parts. That is why the GIGYF2 and RACK1 blots were the same (sorry for not making this explicit in the respective legends). To avoid any concern or confusion, one of the figures (now Figure 4B) has been replaced with a different version of that same experiment.

It is not completely clear how eS10 gets ubiquitinated by ZNF598 in cells missing GIGYF2 (Figure 6). The supplement Figure 6 shows no recruitment of ZNF598 to poly-ribosomes in those same conditions with a rather massive recruitment in case of the WT cells. Authors argue that this could be due to transient interaction of ZNF598 in cells missing either EDF1 or GIGYF2. Is es10 ubiquitinated in HEK293 cells missing ZNF598, or Caco-2 and MCF-7 cells?

We have now verified that the ubiquitination of eS10 is completely dependent on ZNF598 in both wild type cells and in cells lacking GIGYF2 or EDF1 (Figure 6C). Thus, the ubiquitination must be due to transient interactions that do not result in stable co-fractionation through sucrose gradients. Consistent with this conclusion, long exposures of the immunoblots show that although weak, ZNF598 does interact selectively with collided ribosomes even in cells lacking GIGYF2 or EDF1 (Figure 6—figure supplement 2).

Original report by Morita et al., 2012. has 4EHP-GIGYF2-ZNF598 complex as translational repressor with direct co-IPs between all components and especially between 4EHP and ZNF598 as well as GIGYF2 and ZNF598. Is EDF1 binding to GIGYF2-4EHP and competing with ZNF598? There is no co-IP studies shown on any components so the reader is left to believe on interactions through the polysome profiles or functional assays.

We have now added co-IP experiments to our study showing that GIGYF2 interacts with 4EHP and ZNF598, and that EDF1 is not part of this complex away from the ribosome (Figure 5; see detailed reply to the Editor’s comments above).

Reviewer #3The authors should compare their findings with recent work outlining the crosstalk between translation initiation and the RQC: https://www.biorxiv.org/content/10.1101/792994v1

This is now cited at a few key points in the paper. We avoid extensive comparison or discussion as this study has not undergone peer review and might change in the process.

The authors might want to explore the potential sequence/structure features underlying pausing collisions events and their potential in discriminating between ZNF598-dependent and other pausing events. Results from recent work on detecting pausing events should be considered (e.g. https://www.biorxiv.org/content/10.1101/746875v2 , https://www.biorxiv.org/content/10.1101/710061v1 , https://www.biorxiv.org/content/10.1101/710491v1).

As noted in the overall reply to the Editor, we have omitted our own ribosome profiling analysis. Instead, we cite the above studies as evidence for widespread pausing and collisions.

Figure 3C shows how both EDF1 and GIGYF2 migrate to the polysome fraction under emetine treatment. However, this might represent a more general phenomenon where many more protein complexes migrate to the polysome fraction. Can the author comment on this possibility?

This is now clarified in the text where we explain that extensive proteomic analysis shows that the vast majority of the ~600 detected proteins do not show this behavior. Furthermore, of these ~600 proteins, only EDF1 and GIGYF2 showed this behavior in both wild type and ZNF598 cells, which is why these were pursued in downstream experiments. Sorry for not explaining how uniquely specific these two hits proved to be.

[Editors’ note: what follows is the authors’ response to the second round of review.]

The manuscript has been improved but there are some remaining textual issues that need to be addressed before acceptance, as outlined below:1) It was thought that the authors should either tone down some statements or discuss alternative explanations of observed phenomena. In particular, the interpretation that EDF1 recruits GIGYF2 complex at collided ribosomes was found to be insufficiently supported by data. To this end, it was found that although the IP experiments support the role of EDF1 in stabilization of GIGYF2 complex binding to collided ribosomes, their role in its recruitment remains unclear. This should be clearly stated throughout the manuscript and depicted in the model shown in Figure 5. It was also thought that Figure 7 should be removed as this model would require further kinetic studies that, as we agree, are out of the scope of this study.

We have changed the wording in the text to avoid making any claims about EDF1 directly recruiting the GIGYF2 complex to collided ribosomes. Instead, we are cautious to always state that EDF1 “stabilizes” or “facilitates stable engagement” or “facilitates stable binding” of GIGYF2 with collided ribosomes. These are accurate descriptions based on the data: GIGYF2 is selectively associated with collided ribosomes (Figure 1 and Figure 1—figure supplement 1), and this association is impaired in cells lacking EDF1 (Figure 4B; Figure 5C, lanes 3-6). Importantly EDF1 is itself specific for collided ribosomes and this does not depend on ZNF598 (Figure 1) or GIGYF2 (Figure 5—figure supplement 1). Furthermore, a point mutation in EDF1 that impairs its ribosome binding (Figure 4C) is also impaired in translation repression at stalls (Figure 4D), an effect that is mediated by GIGYF2. These findings support the conclusion that EDF1, and specifically its ribosome binding, is needed for effective function of GIGYF2. In the Discussion, we note the two alternative interpretations of these data: (i) a direct model where EDF1 provides part of a binding site for the GIGYF2 complex on collided ribosomes; (ii) an indirect model where EDF1 engagement of collided ribosomes changes the ribosome complex in a way that favours the bound state of the GIGYF2 complex.

We have edited the diagram in Figure 5 to be agnostic to either model. Here, and in Figure 7, we depict all of the factors associated with collided ribosomes rather than showing a hypothetical order of binding. Figure 5 further depicts that off the ribosome, interactions between ZNF598GIGYF2-4EHP are observed, which is supported by the co-IP data in Figure 5 and earlier cited work. In the text our wording is now more cautious: “Although we cannot know the order of binding to the ribosome from these data, we speculate that EDF1 binding may occur first because it is ~50 fold more abundant than either GIGYF2 or ZNF598 (Itzhak et al., 2016).” We are similarly circumspect in the Discussion section on this issue.

We have opted to retain Figure 7 because it is helpful to illustrate for readers what we think is happening. To alleviate the reviewers’ concern, we have now edited the Discussion to better explain our rationale for proposing each aspect of our model. The main point of speculation is that re-starting elongation after an incidental collision is faster than ubiquitination and helicase-mediated ribosome disassembly. We now provide appropriate references and cite specific time ranges for the appropriate reactions. Elongation reactions take between ~200 ms to 4 seconds, while the sum of ZNF598- and ASCC-mediated reactions is likely to take minutes. We are also careful to state that although we have not measured the rates of all binding and enzymatic reactions, our estimates of these are based on sound and well-established biochemical principles. All other aspects of the model are well supported by data, which we will not enumerate again here.

2) Although highly appreciated, several issues were raised regarding frame-shifting studies. For instance, it appears that EDF1 depletion favours -1 frameshifting which cannot be fully explained by the collision model. Moreover, the authors cite studies that indicate that frameshifting should be in +1 direction. Discussion of these apparent discrepancies appears to be warranted. Potential effects of frameshifting on NMD activation should also be discussed. Finally, RFP signal and RFP/GFP ratios should be presented along with GFP for EDF1 knockdown and KO cells (Figure 4 and Figure 4—figure supplement 1) to further facilitate interpretation of results in Figure 6 and Figure 6—figure supplement 3.

The mechanistic basis of frameshifting at collisions is not known. At this point, it remains a correlative observation, both in our study and in two earlier studies. The earlier studies used different reporters, a different organism (yeast), and different means of stalling the ribosome. The mechanism of stalling is clearly influenced by organism (since di-codons or stem-loops that stall in yeast have little or no effect in mammals), so direct comparisons between yeast and mammals have not been possible. Importantly, the mechanism of stalling seems to influence frameshifting: di-codons seem to prefer +1, whereas a stem-loop permits both -1 and +1 (Wang et al., 2018). Thus, on the basis of earlier studies in yeast, there was no reason to necessarily expect any particular result in mammalian experiments that use qualitatively different mechanisms of stalling. Thus, there are no discrepancies; simply a set of difficult-tocompare observations using different reagents, methods, and organisms to probe a poorly understood phenomenon. We now state that our results “roughly approximate” recent observations in the yeast system, then explicitly highlight the above issues.

We also mention that an appreciable role for NMD can be excluded because the GFP signal does not change with the frameshifting constructs despite clearly showing premature termination (i.e., the RFP signal is markedly reduced). This is consistent with past work showing that in mammals, a downstream exon junction complex is typically needed to trigger NMD, so it would not be expected for an intronless reporter. Finally, the requested individual GFP, RFP, and ratio plots of the experiments in Figure 4 are provided as a new supplement to Figure 6 so they can be compared by any interested reader (Figure 6—figure supplement 4).

3) It was also thought that the authors should specify number of replicates and precise criteria for some statistical tests. For example, 1.6 fold threshold was found to be quite arbitrary, and the authors should clearly elaborate on criteria to use such cutoff. MS data should be made publicly available and identified factors that are enriched/depleted on collided ribosomes should be at least briefly discussed. The authors are are also encouraged to add all appropriate controls (e.g. showing that ZNF598 is indeed knocked out in the experiment in question rather than referring to previous publications where these lines were established)

The number of replicates as well as other important information related to data acquisition and analysis for each experiment is available in the Materials and methods section. The complete quantitative mass spectrometry data (as an annotated Excel file) is provided with the paper, just as it was provided with the initial submission and the revision. The raw files have been uploaded to the PRIDE database by our mass spectrometry facility. Dotted lines indicating 3 standard deviations from the mean have been added to the graphs in Figure 1 to provide a point of reference. EDF1 and GIGYF2 were chosen for further study because they are the two most collision-specific proteins in both wild type and ZNF598 knockout cells. This reason is stated in Results paragraph one. A cut-off or threshold is irrelevant, since our choice to study these was because they were the most enriched, which would be the case regardless of any chosen threshold.

Other proteins that we did not study further are not discussed because we have nothing productive to say about them. We only labelled them on the graph in Figure 1A because they were all provisional candidates until we did the secondary analysis in Figure 1B that highlighted EDF1 and GIGYF2. Because we provide the full dataset, any interested colleague can read about them and pursue them as desired.

The absence of ZNF598 in knockout cells is documented in Figure 1 and in the MS dataset provided, where it is preferentially detected in wild type, but not knockout cells. This should alleviate any concern that the cell line has somehow reverted since our previous publication, which is also clear from its characteristic phenotype (Figure 3C and Figure 6—figure supplement 3). The knockdowns and knockouts of all other components are documented with immunoblots within the paper.

4) At times, interpretation of data was thought to require some clarification. Particularly, Figure 2—figure supplement 1A shows apparent increase in mRNA levels, albeit in the text the authors state that there is a lack of the potential mRNA abundance changes. To this end, it was thought that appropriate significance testing is required to corroborate the authors' conclusions, and if it turns out that the changes in mRNA abundance are significant, this should be commented on. Moreover, it is argued that 4EHP, but not GIGYF2 depletion increases translation. However, considering the scale in Figures 2C vs. 2D, it appears that the changes are of comparable amplitude but in the opposite directions. Based on this, significance testing appears warranted.

No differences are seen in mRNA levels by statistical comparisons, and this is stated in the legend. Regarding the second point, the reviewer appears to have mis-interpreted the data, perhaps by overlooking the fact that Figure 4C, 4D, and 4E are performed in different genetic backgrounds. Both GIGYF2 depletion and 4EHP depletion each increases translation in wild type cells (Figure 4C). Figure 4D documents that 4EHP depletion does not increase translation further in GIGYF2 knockout cells, providing genetic evidence that they act in the same pathway. Figure 2E documents that GIGYF2 depletion and 4EHP depletion each increases translation in ZNF598 knockout cells, illustrating that ZNF598 is not required for this effect.

5) The authors are advised to verify some of the references. For instance in Mills et al., 2016 study does not appear to demonstrate ZNF598 loss during maturation of reticulocytes.

Mills et al. did not study ZNF598 specifically, but they do provide ribosome profiling datasets of reticulocytes. Inspection of this dataset reveals that footprints corresponding to ZNF598 are not detected in reticulocytes. Importantly, footprints of other mRNAs that are similar in abundance to ZNF598 in a pre-reticulocyte cell line are detected in reticulocytes. This supports the idea that ZNF598 mRNA is reduced or absent in reticulocytes, consistent with the absence of ZNF598 protein in reticulocyte lysate as documented in Juszkiewicz et al., 2018, which is cited together with Mills et al., 2016.